# Tough soldering for stretchable electronics by small-molecule modulated interfacial assemblies

Liqing Ai [1,5], Weikang Lin [2,3,5], Chunyan Cao [1,5], Pengyu Li [2,3], Xuejiao Wang [1], Dong Lv [1], Xin Li [1], Zhengbao Yang [2,3] ✉ & Xi Yao [1,4] ✉

The rapid-developing soft robots and wearable devices require flexible conductive materials to maintain electric functions over a large range of deformations. Considerable efforts are made to develop stretchable conductive materials; little attention is paid to the frequent failures of integrated circuits caused by the interface mismatch of soft substrates and rigid silicon-based microelectronics. Here, we present a stretchable solder with good weldability that can strongly bond with electronic components, benefiting from the hierarchical assemblies of liquid metal particles, small-molecule modulators, and non-covalently crosslinked polymer matrix. Our self-solder shows high conductivity (>2×10$^5$ S m$^{-1}$), extreme stretchability (~1000%, and >600% with chip-integrated), and high toughness (~20 MJ m$^{-3}$). Additionally, the dynamic interactions within our solder's surface and interior enable a range of unique features, including ease of integration, component substitution, and circuit recyclability. With all these features, we demonstrated an application as thermoforming technology for three-dimensional (3D) conformable electronics, showing potential in reducing the complexity of microchip interfacing, as well as scalable fabrication of chip-integrated stretchable circuits and 3D electronics.

Stretchable electronics with high toughness and reliable electrical stability are essential in the applications of soft robotics[1], wearable electronics[2,3], and on-skin electronics[4,5]. Substantial progress has been made in developing stretchable polymers with intrinsic conductivity as well as flexible nanocomposites with conductive fillers for stretchable sensors[6] and integrated circuits[7]. However, most conductive circuits often lose their function under strains that are much lower than their stretching capability, due to the incompatibility between conductive domains and neighboring polymer networks[8-10]. In particular, the interface mismatch between soft substrates and rigid microelectronics[11,12] results in irreversible changes in electrical resistance and permanent interface failure between conductors and components[13,14]. It is therefore a high demand to develop stretchable solders that can bind with rigid electronic components and maintain reliable conductivity over a large range of strains.

Gallium-based liquid metals (LM) with a unique combination of metallic conductivity and fluidity at room temperature stand out as a competitive conductive filler in the development of stretchable conductive materials[15,16]. Recently, LM-polymer composites with large stretchability and high electrical conductivity have been developed[17,18]. While the weak interactions between the LM and polymer substrates usually cause undesired leaking upon repeated stretching, which deteriorates the mechanical properties and conductive stability of these conductors[19]. To mitigate these issues, surface engineering

[1]Department of Biomedical Sciences, City University of Hong Kong, Hong Kong 999077, China. [2]Department of Mechanical & Aerospace Engineering, Hong Kong University of Science and Technology, Hong Kong 999077, China. [3]Department of Mechanical Engineering, City University of Hong Kong, Hong Kong 999077, China. [4]City University of Hong Kong Shenzhen Research Institute, Shenzhen 518000, China. [5]These authors contributed equally: Liqing Ai, Weikang Lin, Chunyan Cao. ✉e-mail: zbyang@ust.hk; xi.yao@cityu.edu.hk

technologies such as chemical surface modification and interfacial manipulations have been employed[20–22]. However, it is still challenging to integrate LM composites with rigid electronic components because of the fluidity and poor adhesion of LM. There are a couple of pioneering attempts showing potential of solving these interfacial issues, such as adding additional encapsulation layers[23], providing proper barriers[24] for generating interlocking effect between LM and components, or adding interface binder to make temporary connections by virtue of physicochemical bonding[25,26], but these methods cannot form a robust interface between the soft conductive composites and rigid electronics directly.

Supramolecular polymers have been demonstrated as a promising platform to carry conductive fillers such as silver nanowires and carbon nanotubes to form multifunctional conductive composites for stretchable electronic and wearable devices[27,28]. Their advantageous mechanical and interfacial properties benefiting from versatile design of non-covalent bonds and the resulted dynamic networks may provide a unique approach to solving the aforementioned challenges[29]. We here report the fabrication of LM-based supramolecular polymeric composites with high mechanical toughness, high conductivity, and robust electrical connections with electronic components. The composites are prepared by the co-assembly of a molecular engineered supramolecular polymer, and small-molecule modulators stabilized LM microparticles (LMP). The dynamic interactions within the surface and interior of the composite enable the LM-polymer composite as a self-solder to interconnect with commercial silicon-based electronic

components in integrated circuits through a typical thermal processing method (Fig. 1a–d). During the soldering process, the ULPC deforms to better wrap the mounted resistor after thermal processing (Fig. 1e). The dynamic interactions within the ULPC enable a stable interconnect between both electronic components and TPU substrate, to withstand the significant shear stress generated during the thermoforming process. Notably, the circuit (Fig. 1f) exhibits exceptional stability and robustness without requiring encapsulation, which simplifies component substitution and circuit recyclability.

## Results

### The design principle and polymer synthesis

The dispersion status of LMP in polymer matrix and their interfacial interactions together determine the comprehensive performance of the LM-polymer composites[30,31]. Although various fabrication methods have been reported in the development of LM-polymer composites[32,33], huge challenges exist in the regulation of interfacial interactions between LMP and polymer matrix[34,35]. To overcome the limitation, we use a molecular engineering strategy to synthesize a typical linear polymer and small-molecule modulators, both of which are functionalized with terminal 2-amino-4-hydroxy-6-methylpyr-imidine (UPy) motifs (Fig. 1b–d). Compared with traditional designs which LMP are simply blended and sintered within polymer matrix, the use of small-molecule modulators can bring a couple of unique advantages. Chemical motifs that show high affinity to the oxide layer of LM can be precisely incorporated in the small-molecule modulator

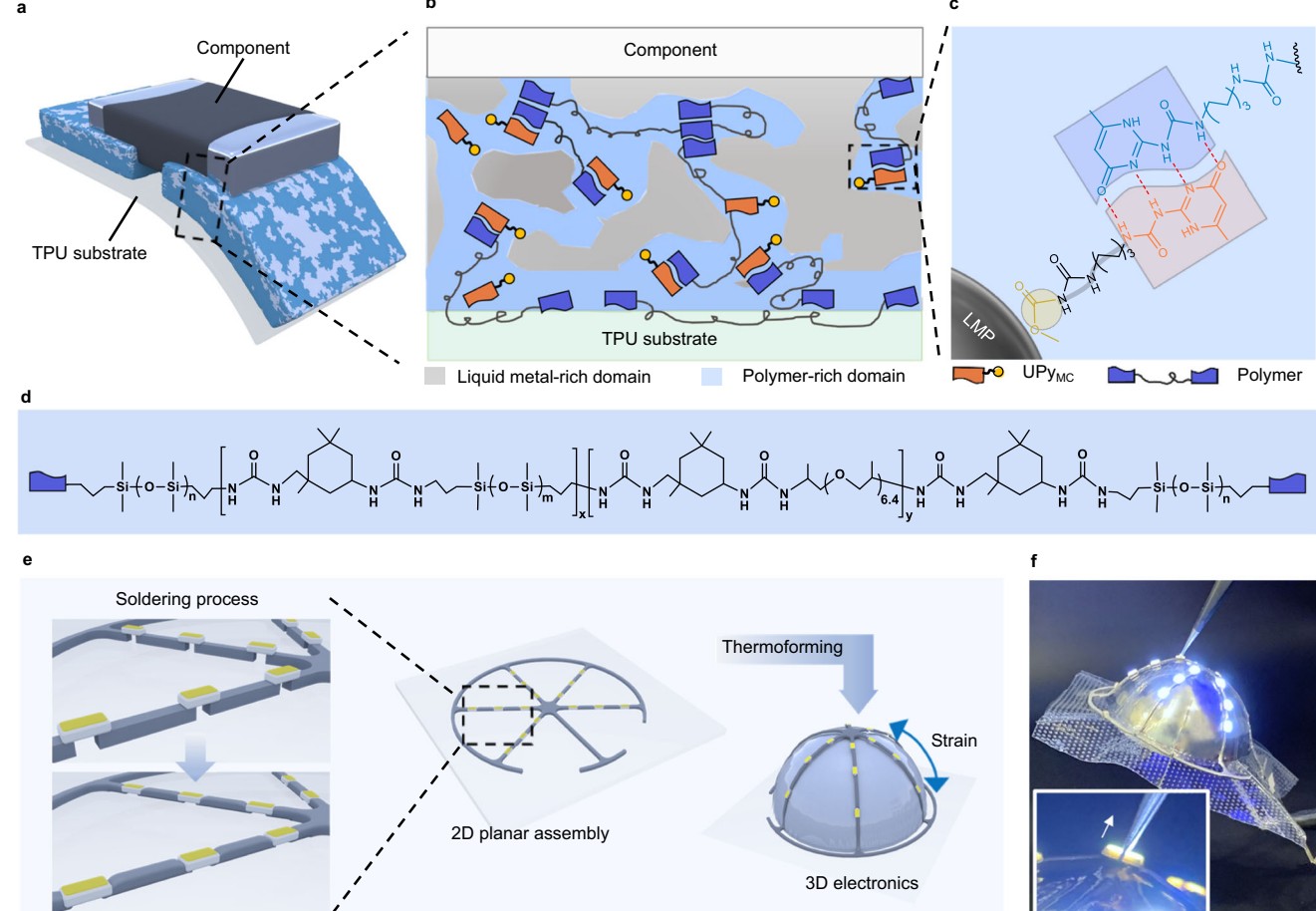

**Fig. 1 | Schematic of the soldering and thermoforming process. a** Schematic illustration of the LM-based composite design and its integration with TPU substrate and electronic component. **b** Schematic illustration of the hierarchical assembly within the composite. **c** Molecular interactions between the polymer, LMP, and UPy$_{MC}$ modulators. **d** Molecular structure of the synthesized polymer. **e** Schematic illustration of the soldering and thermoforming processes. **f** Optical image of a 3D electronic enabled by the ULPC. Inset: The LED remains lit even under the action of external force lifting the entire electronics (50 g).

to facilitate LMP generation. The small-molecule modulator can also be inserted into polymer coils, regulating the assembly of polymer chains through hydrogen bonding interactions. The capacity of molecular stacking and aggregation of UPy motifs can assist the formation of dynamic network within the polymer matrix which in return stabilizes the compartmented LM. All these together result in a hierarchical assembly for which a heterogeneous LM-polymer interface can be stabilized.

The mechanical properties particularly the stretchability and toughness of the polymer matrix are key for their applications in stretchable electronics. We synthesize the supramolecular polymers through typical condensation reactions between amino terminated monomers, UPy precursors, and diisocyanate linkers (Supplementary Fig. 1). An aliphatic extender is included to tune the chain length and adjust the density of hydrogen bonding motifs. Terminal UPy motifs are incorporated into the polymer main chains to help build intermolecular hydrogen bonding[36]. A low density of UPy motifs is preferred as this can avoid their cascade crystallization[37,38], and facilitate the stretchability of the polymer[39]. Polydimethylsiloxane (PDMS) segments are incorporated in the main chain, because of the low surface energy and low glass transition temperature[40]. The molecular weights (MWs) of UPy-terminated PDMS segments are tuned at ~1000, ~2000, and ~3000, and the feeding amounts of UPy monomers are varied from 0.02, 0.04 to 0.08 (equivalent to the molar fraction of the UPy units relative to total amine groups) for comparison and optimization. With the parameters set above, the stretchability (392–1986%) and Young's modulus (0.4–34 MPa) of the synthesized polymer samples are highly tunable (Supplementary Figs. 2 and 3, Supplementary Table 1), and the toughness varies in a narrow range (between 9.2-11.3 MJ m$^{-3}$), demonstrating an effective platform for the subsequent fabrication of LM-polymer composites.

## Small-molecule modulated hierarchical assemblies

The UPy-functionalized small molecule modulators play a key role in improving the interaction between the LMP and the synthesized polymer. They are incorporated with functional groups such as ester and carboxylic acid groups, which are proven to show high affinity to oxide layer of LMP[41,42]. As a typical example, ester functionalized UPy modulators are synthesized (termed UPy$_{MC}$, [1]H NMR see Supplementary Fig. 4). We then fabricate UPy$_{MC}$ modulated LM-polymer composite (termed ULPC) by a typical sonication-assisted solution processing method. UPy$_{MC}$-wrapped LMP (termed UPy$_{MC}$-LMP) is prepared by ultrasonic treatment of UPy$_{MC}$ dispersion and bulk LM. ULPC precursor is obtained by mixing the UPy$_{MC}$-LMP suspension with polymer solution (Fig. 2a). During solvent evaporation, the LMP will precipitate and transform into heterogeneously structured LM-rich layer in the final ULPC films (Supplementary Fig. 5).

The structure and composition of UPy$_{MC}$ stabilized LMP are evaluated via energy-dispersive X-ray spectroscopy (EDS), transmission electron microscopy (TEM), and X-ray photoelectron spectroscopy (XPS). As shown in Fig. 2b, the UPy$_{MC}$ can adsorb on the surface of LMP and form a dense wrapping layer as revealed by the TEM images. EDS analysis revealed a homogeneous distribution of carbon elements surrounding the LMP surface, providing further evidence of the uniform wrapping of UPy$_{MC}$ around the LMP surface (Supplementary Fig. 6). XPS spectrum revealed that the intensity and position of three representative peaks for -NH (398.7 eV), -N-C (397.9 eV), and -N=C (399.4 eV) groups of UPy$_{MC}$ changes substantially in the UPy$_{MC}$-LMP[35], implying strong interactions between the UPy$_{MC}$ and the surface of LMP (Fig. 2c).

As addressed previously, small-molecule modulators are expected to co-assemble with the UPy-terminated polymers through hydrogen bonding interactions. Therefore, the assembly status of the ULPC is carefully studied. Fourier-transform infrared spectroscopy (FTIR) shows the typical UPy characteristic peak of C=O stretching in

ureido at around 1698 cm$^{-1}$, while the corresponding peak in the polymer is around 1703 cm$^{-1}$ due to the linking with PDMS molecular chain[43] (Fig. 2d). The peak broadening at 1698–1703 cm$^{-1}$ in the ULPC suggests that the self-assembly between UPy$_{MC}$-LMP and polymer has been successfully constructed[35]. The semi-crystalline property of the synthesized polymer can be confirmed by X-ray diffraction (XRD) (Fig. 2e). Compared with the pristine polymer sample, the polymer-UPy$_{MC}$ sample shows higher crystallinity due to the higher molar ratio of UPy motifs which can contribute to further aggregation and stackings. Similarly, the ULPC sample shows reduced crystallinity because the addition of LMP is expected to dilute the overall polymer ratio. Wide-angle X-ray scattering spectra is used to study the microscale crystalline properties of the samples. The size of crystalline domains changes very little after adding LMP (Supplementary Fig. 7), indicating that LMP has negligible effect on the molecular aggregation and crystalline structure at microscale, which is expected to enhance the mechanical properties without scarifying the stretchability of the polymer composite[44,45].

## High toughness and high stretchability of ULPC

The unique hierarchical assembly and crystalline properties of the ULPC will affect the mechanical properties substantially. The mechanical properties of the pristine polymer film, polymer-LM film, polymer-UPy$_{MC}$ film, and as-prepared ULPC are studied via tensile tests (Fig. 2f). Notably, the direct incorporation of high mass ratio of LMP (30%) into polymer leads to a sharp deterioration of mechanical strength due to the aggregation of LMP as previously reported. While for the ULPC sample, the tested tensile strength and toughness are enhanced remarkably (Supplementary Table 2). Compared with polymer-LM film, the resulting ULPC film exhibits a 450% and 370% increase in both the ultimate tensile strength (3.2 MPa) and toughness (19.8 MJ m$^{-3}$), respectively. The polymer-UPy$_{MC}$ sample also shows higher tensile strength, while the elongation at break has a remarkable decrease due to the high density of hydrogen-bonding crosslinks and the high crystallinity. Addition of 10 wt% of LM can improve the stretchability from 480% to 800%. Further increment of the LM content from 10 wt% to 40 wt% would further increase the breaking elongation, while the tensile stress decreases accordingly (Supplementary Fig. 8).

To further understand the role of UPy modulators in the regulation of the interfacial interaction between the polymer and LMP as well as the mechanical enhancement of the composite, we first test mechanical properties of ULPC samples with different amounts of UPy$_{MC}$ modulators. For those samples, the introduction of ~1 wt% UPy$_{MC}$ can impressively improve the toughness of the polymer-LM composite from 5.3 to 15.3 MJ m$^{-3}$ (Fig. 2g, Supplementary Table 2). Further increment of the UPy$_{MC}$ fraction from 2 wt% to 6 wt% would cause the reduction of the breaking elongation, although the tensile stress can increase accordingly. To further confirm the role of UPy modulators, another three types of UPy modulators with different chemical structures are synthesized for the preparation of ULPC samples. Two of them are similar to UPy$_{MC}$, but the terminal methyl ester groups are changed to carboxylic acid (UPy$_{Gly}$) and propyl groups (UPy$_{BI}$). The UPy$_{Gly}$ is expected to show similar interactions with LM, while the UPy$_{BI}$ is expected to show limited interactions with LM. The Glycine (Gly) molecule is selected as the third candidate to clarify the function of UPy motifs (Structures see Supplementary Fig. 4). Tensile test shows that the strength of the sample prepared by UPy$_{Gly}$ is much higher than that of other samples, indicating the interactions with both LMP and polymer are necessary for the mechanical enhancement (Fig. 2h). To further elucidate the role of polymer matrix in the mechanical enhancement, we change the content of UPy monomers in the pristine polymer (Supplementary Fig. 9, Supplementary Table 3). It is observed that the enhancement effect is proportional to the UPy content in the polymer, which is also strong

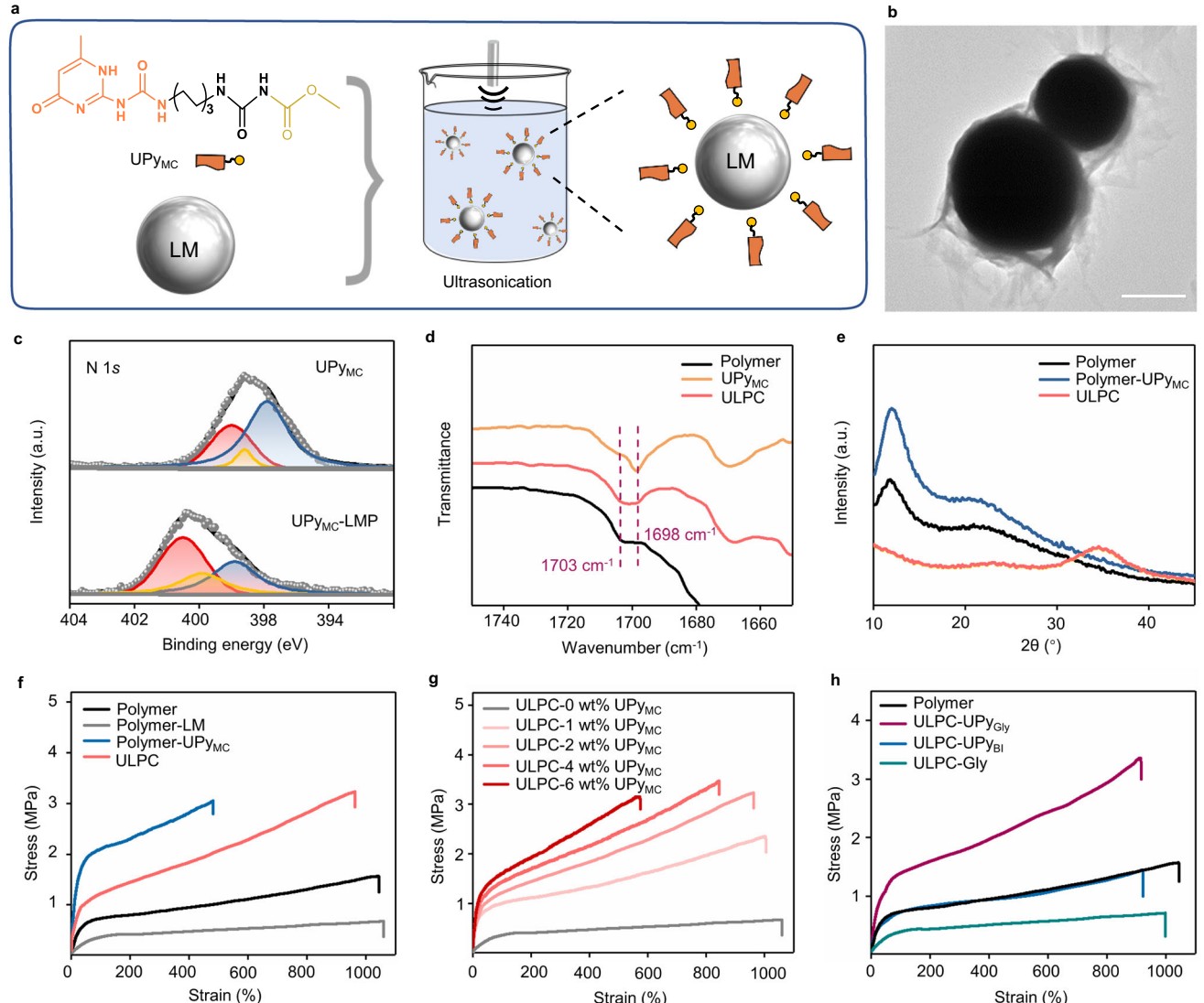

**Fig. 2 | Characterization of the UPy$_{MC}$-LMP and the ULPC solder. a** Schematic illustration of the generation of UPy$_{MC}$-LMP by ultrasonic treatment of bulk LM and UPy$_{MC}$ modulators. **b** TEM image of the as-prepared UPy$_{MC}$-LMP, scale bar, 1 μm. Each experiment was repeated independently 3 times with similar results. **c** XPS of the N 1$s$ of UPy$_{MC}$ modulators and UPy$_{MC}$-LMP. Shifts are observed for -NH/-N-C/-N=C motifs in the particles. **d** FTIR spectra of UPy$_{MC}$, polymer, and ULPC. **e** XRD spectra of polymer, polymer-UPy$_{MC}$, and ULPC. Comparison of the tensile stress-strain curves for the polymer and composites prepared by **f** different compositions, **g** different weight ratios of UPy$_{MC}$, and **h** different types of small-molecule modulators. (Except where stated otherwise, polymers were prepared from siloxane oligomers with ~2000-MW and 0.08 amount of UPy monomers, and ULPC was fabricated from the polymer with 30 wt% LM content and 2 wt% UPy$_{MC}$).

evidence that intermolecular interactions between the UPy-functionalized modulators and polymer matrix are responsible for the enhancement.

## Interface-dependent electromechanical properties of ULPC

As shown in Fig. 3a, the surface of ULPC shows a heterogeneously structured LM-rich layer. The scanning electron microscopy (SEM), its corresponding energy dispersive X-ray (EDX) element mapping, and Raman mapping of LM-rich layer confirmed the hybrid surface, and the surrounded polymer compartments tightly retain LM to avoid leaking (Supplementary Figs. 10 and 11). To demonstrate the robust interactions between the LM and the ULPC samples, we following evaluate and compare the shape evolution of LM droplets on the surface of pristine polymer and ULPC films. As the sample film is stretched under continuous tensile strain from 200% to 500%, the boundary of LM droplet extends from ~165% to ~320% simultaneously when it is deposited on the pristine polymer, while the boundary of LM droplet extends from ~195% to ~495% on the ULPC samples, respectively. After

the removal of the oxide layer of LMP through HCl treatment, the LM droplet cannot deform with the stretching of the polymer composite. These results prove the strong and adaptive interactions between the ULPC and the oxide layer of LMP (Supplementary Fig. 12).

Micro-computed tomography (Micro-CT) images are conducted to get more insight of interior structure - LMP tend to aggregate and form continuous percolating paths within the composite, which is essential for the electrical stability of the conductive materials[46] (Fig. 3b and Supplementary Fig. 13). With the compact assembly of LMP, peeling films of ULPC can generate sufficient stress to percolate particles for electrical activation[47] (Supplementary Fig. 14). Since the LM content in the hybrid interface contributes to the conductivity and the polymer content contributes to the interfacial adhesion, such unique hybrid interface is beneficial for interfacial contact with high conductivity and high adhesion[9]. We further developed a series of composites with different weight ratios of LM in the composite to study the conductivity and the interfacial adhesion. We note that it may be intuitive that an increase in LM concentration would facilitate

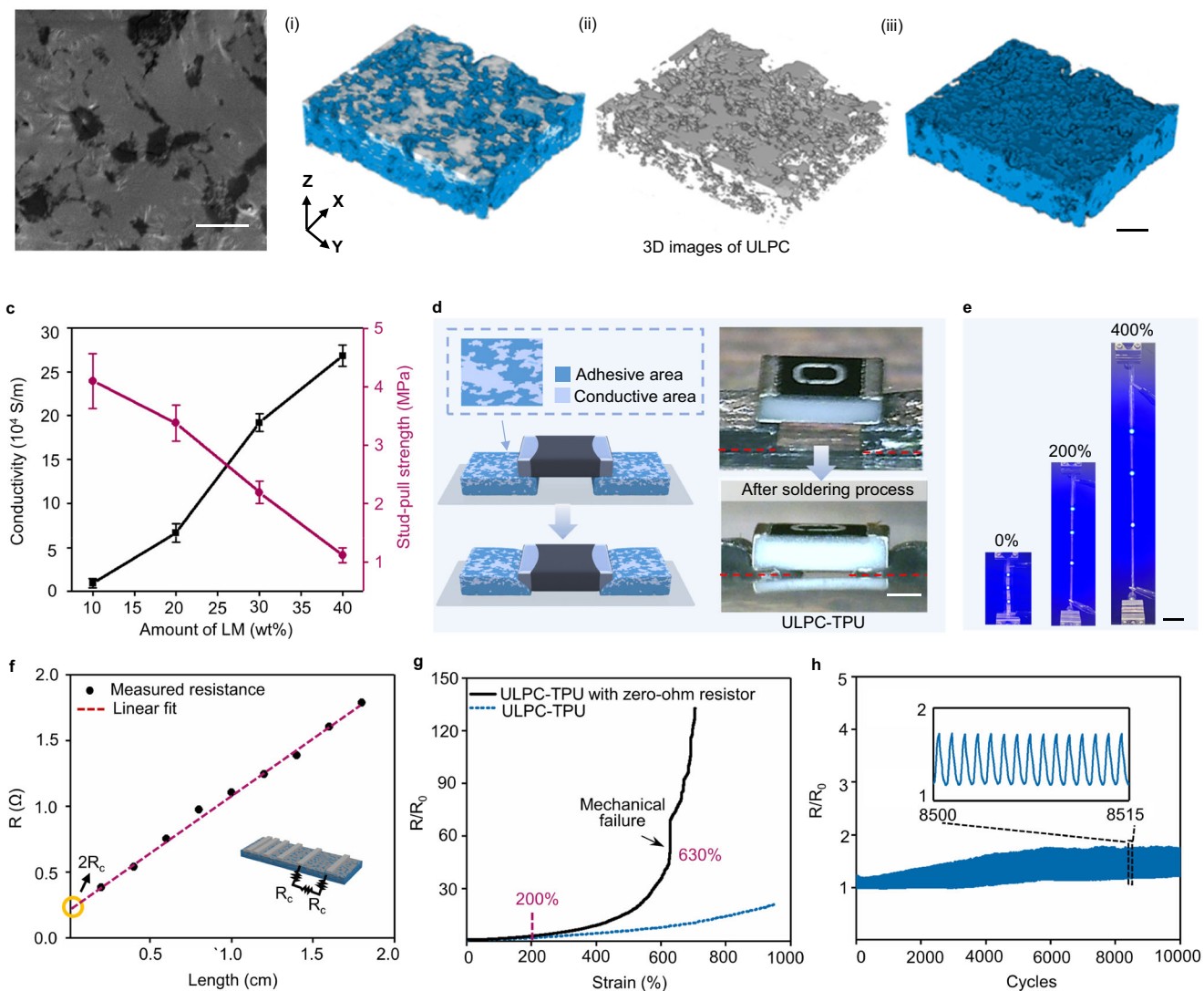

**Fig. 3 | Interface-dependent electromechanical properties of integrated ULPC.** **a** SEM image of the surface of ULPC LM-rich layer, scale bar, 10 μm. Each experiment was repeated independently 3 times with similar results. **b** Reconstructed 3D Micro-CT images of i) the LM-rich layer, ii) LM only, and iii) polymer only, of a representative ULPC sample, scale bar, 50 μm. **c** The measured conductivity and maximum stud-pull strength of the ULPC with different weight ratios of LM. All values represent the mean ± SD for $n = 3$ independent experiments. **d** Schematic illustration and optical images of side view of the soldering process of ULPC-TPU with a zero-ohm resistor, scale bar, 1 mm. **e** Photographs of ULPC-TPU with LED array under strains, scale bar, 10 mm. **f** Contact resistance measurement for a ULPC interface using the transmission line method (half of the y intercept: $R_c = 0.11$ Ω, $R^2 = 0.9961$). The inset shows the schematic of the contact resistance measurement. **g** Comparison of relative resistance change of ULPC-TPU over strains with or without an embedded zero-ohm resistor. **h** Resistance stability of ULPC-TPU in 10000 stretch-release cycles at 50% strain.

electrical conductivity (from 10000 to 250000 S m⁻¹); however, the maximum stud-pull strength would accordingly decrease from 4.1 to 1.1 MPa (Fig. 3c). Since the 2.2 MPa adhesion (competitive with commercial copper tape) is sufficient to meet the requirements in most practical scenarios, the ULPC with LM loading of 30 wt% is investigated for the application of stretchable electronics. We also test its adhesion by 90° peel-off test and shear test to show the robust integration of ULPC with components (Supplementary Figs. 15 and 16). As a comparison, the LM-polymer composite films without adding UPy$_{MC}$ show poor conductivity and limited stretchability before electricity failure (initial resistance >200 Ω, Supplementary Figs. 17 and 18), which again illustrates that UPy$_{MC}$ can largely improve the interactions between LM and polymer substrate.

The ULPC is cut into stripes at 1.3 mm width and different lengths by a cutting machine and then mounted on commercial Thermoplastic polyurethane (TPU) film through thermal release tape-assisted thermal

transfer processes (See methods for details). Accordingly, various electronic components can be integrated to the resulting ULPC/TPU composite film (termed as ULPC-TPU). ULPC-TPU shows reduced stress-strain hysteresis in the cyclic tensile tests at 100% strain region compared with bare ULPC films (Supplementary Fig. 19). We further perform in situ peeling tests to reveal the interface stability of the ULPC sample. It is noteworthy that the ULPC exhibits marginal changes in resistance when Scotch tape was repeatedly peeled off from the conductive interface, as illustrated in Supplementary Fig. 20. Specifically, small resistance changes (~1%) are observed when the Scotch tape was attached on the ULPC sample, but the resistance promptly returns to its original value when the tape was peeled off. This demonstrates the excellent electrical stability of the robust interface.

In addition to stable electrical resistivity, stable adhesion with electrical components is also a crucial property of stretchable solder. The stable adhesion relies on molecular bond exchange and

recombination when the ULPC is brought into contact with electrical components[27]. Welding of the ULPC at temperatures slightly above the glass transition temperature (Tg) is more favorable due to the enhanced molecular bonding ability[48]. The Tg of ULPC is measured at 113 °C (Supplementary Fig. 21), lower than many commercial flexible polymers, indicating well rheological weldability and good compatibility with thermal processing (See methods for details).

To investigate the electromechanical properties of ULPC-TPU with electronic components, we attach a zero-ohm surface-mount resistor on the ULPC through soldering processing to form a basic but representative circuit. Figure 3d shows schematics and optical images of a resistor-mounted ULPC-TPU, where the polymer composite deforms to better wrap the mounted resistor after thermal processing[49]. The ULPC leverages dynamic interactions to establish a stable interconnect between both electronic components and TPU substrate. To visualize the stable electromechanical properties of the ULPC with multiple components, a basic in-series circuit with three LED is stretched to 400% before breaking (Fig. 3e). After soldering process, we used the transmission line method[50] to measure the contact resistance between ULPC-TPU and the zero-ohm resistor, and the calculated contact resistance is $0.11\,\Omega$ (Fig. 3f). Besides, the recorded breaking strain of the integrated circuit is at ~700% (Supplementary Fig. 22), a value 6 times larger than previous study that demonstrated integrated circuits without encapsulation layers[26], and as good as encapsulation circuits[51].

As shown in Fig. 3g, the resistance change of the integrated circuit shows a similar trend with ULPC-TPU under 200% strains ($R/R_0 < 5$). Until the strain up to 600%, a visible increase of resistance change is observed, due to the mechanical failure on the interconnect between ULPC and electronic components (Supplementary Fig. 23). Moreover, the resistor-mounted ULPC-TPU exhibits excellent durability, maintaining its functionality for over 10000 repeated cycles at a loading strain of 50% and 2500 repeated cycles at a loading strain of 100% (Fig. 3h and Supplementary Fig. 24). ULPC-TPU has outstanding performance in its extreme stretchability (~1000%), small resistance changes over a large range of strains, and strong adhesion, the performance comparison with state-of-the-art LM-based stretchable conductive composites are illustrated in Supplementary Table 4.

## Applications in three-dimensional conformable electronics

Three-dimensional (3D) electronics with customizable features, conformability, and stretchability are in high demand for wearable electronics. Thermoforming, a cost-effective and scalable manufacturing method, is often used to create 3D-shaped electronics. However, thermoforming requires conductive materials that can maintain conductivity under high temperatures and stretching. Additionally, the strong shear force between flexible conductors and rigid electronic devices during the thermoforming process can cause electrical components to fracture, leading to electrical failure.

Here, leveraging the high conductivity and strong adhesion with both electronic components and TPU substrate, we fabricate circuit in 2D planar using cost-effective commercial technology. Subsequently, we employ thermoforming techniques to create complex and functional 3D wearable electronics. The ULPC-based 3D electronics are produced by following the steps illustrated in Fig. 4a. The circuit fabrication process typically includes four steps: including mechanical cutting, thermal transfer, pick and place, and thermoforming. To create the desired circuit, the ULPC film is patterned using a cutting machine. The patterned ULPC circuit is then transferred to a TPU substrate by thermal release tape. The thermal release-assisted transfer process is crucial for maintaining the precise position of the ULPC stripes which is essential for accurate electronics welding because the conductive track need to conform to the size of chips and other electronic components. Once the electronic components are fixed in 2D planar, they can be molded into various 3D structures using

thermoforming techniques. For example, a 3D electronic circuit with fifteen light-emitting diodes (LEDs) on the half sphere and their interconnection via ULPC electrodes is made as presented in Fig. 4b. The ULPC-based 3D electronic is reliable under stretching, twisting, and folding deformations without electrical disconnection (Fig. 4c). Conversely, the silver paste displayed notable issues including cracking and inadequate adhesion, ultimately resulting in interface failure with the LED after thermoforming (Fig. 4d).

In contrast to previous works, no encapsulation layer is needed to cover the circuit for components fixation, which significantly facilitating chips and components replacement. Similar to many thermoplastic elastomers, the maximum stud-pull strength of ULPC-TPU gets reduced significantly at sub-zero temperatures due to the rigid characteristics of the polymer[52] (Supplementary Fig. 25). Therefore, we cool the circuit to −10 °C for 20 minutes to peel off the components and substitute with new ones after a new round thermal processing. The integrated circuit demonstrates robust reusability as evidenced by minimal changes in resistance and maximum stud-pull strength throughout repeated peeling-substitution cycles (Supplementary Fig. 26). Furthermore, we demonstrate the continuous substitution of LED components, which can be verified by the well-lit LED after substitution (Fig. 4e). Compared with commercial copper tape, the adhesion performance of ULPC remains stable in the repeated peeling-substitution cycles (Supplementary Fig. 27). Since the absence of permanent crosslinks in the ULPC, the whole circuit can be easily disassembled and recycled into its constituent elements. This recycling process can be accomplished with just a few simple steps, preserving conductivity and quality without significant degradation (Fig. 4f, g, Supplementary Fig. 28).

To demonstrate the potential of ULPC-TPU 3D electronics in the high resolution and complex 3D circuit, which allows conformal and intimate interactions with skins for wearable applications. As shown in Fig. 5a, we have developed a specially designed intelligent heating electronic patch that is suitable for the complex surface of the knee joint, which exhibits non-zero Gaussian curvature. The circuit with chips and a variety of electronic components enables users to control thermal stimulation via a capacitive touchpad (Fig. 5b and Supplementary Figs. 29 and 30). The large area ULPC-based heater shows consistent and low resistance, allowing for rapid temperature responses and operation at a low voltage (Fig. 5c). The minimal temperature fluctuations observed during stretching can be attributed to the negligible resistance changes achieved through the implementation of the serpentine Peano curve design (Fig. 5d). Infrared camera images of the heater exhibit uniform surface temperature distributions across various bending angles of the knee joint (Fig. 5e). The conformal attachment of the electronic patch on the skin not only enhances comfort but also ensures reliable heat transfer for thermal therapy.

In addition, we have expanded the application of our solder with other circuits (rigid, flexible, or stretchable) as a stretchable connector. By soldering ULPC to flat flexible cables and flexible silver conductors, we have successfully formed soft-rigid and soft-soft stretchable connections (See Supplementary Fig. 31 for details). These substrates are compatible with printed circuit board manufacturing, allowing the full utilization of the performance of chips and electronic components. This demonstration showcases the potential of our solder in building stretchable hybrid devices with different functionality and complexity.

## Discussion

In this study, we design and synthesize linear polymers and small-molecule modulators with UPy motifs, which can co-assemble with LMP to produce tough and weldable composites. The hierarchical assembly facilitated by small-molecule modulators can enhance the interaction between the polymer matrix and the LMP, and provide unique interface-dependent electromechanical properties of the resulting ULPC, which are beneficial for high conductivity and high

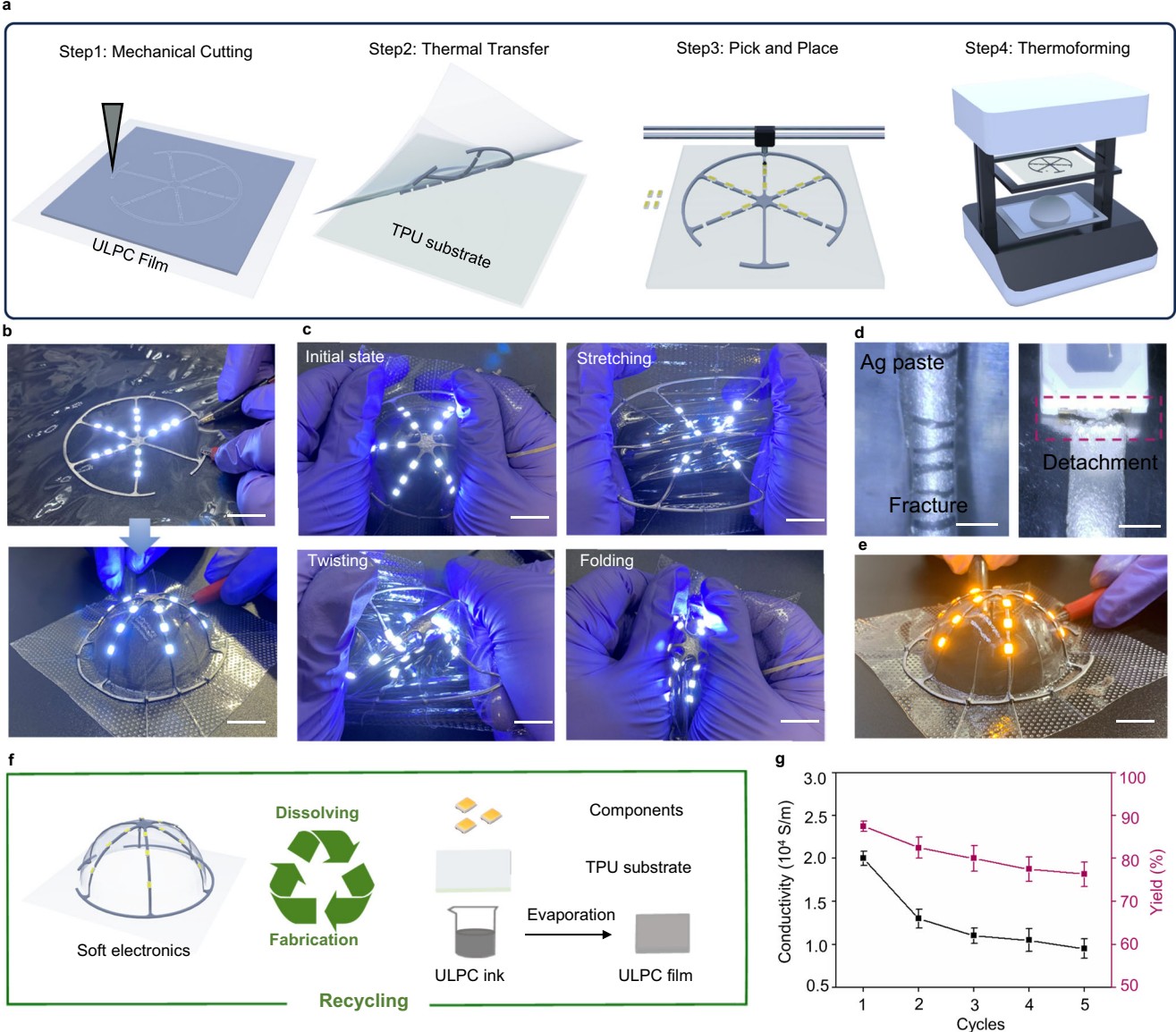

**Fig. 4 | Development of the integrated circuit for 3D electronics. a** Fabrication process for integrated 3D circuit. **b** Images of the integrated process, scale bar, 5 mm. **c** Images of the integrated 3D circuit under stretching, twisting, and folding, scale bar, 5 mm. **d** Optical images of commercial Ag paste after thermoforming, scale bar, 1 mm. Each experiment was repeated independently 3 times with similar results. **e** Optical image of the circuit cooled to −10 °C, peeling off and substituting LED array, scale bars, 5 mm. **f** Recycling process: the circuit is first placed into ethanol, and it can be de-assembled to constituent elements and recycled. **g** The measured conductivity and yield of the ULPC over repeated recycling operations. Here, yield is defined as mass after recycling divided by initial mass. All values represent the mean ± SD for n = 3 independent experiments.

adhesion at the interface contact. We show a record-breaking maximum strain of >600% before electrical failure and relatively small resistance changes over a large range of strains (20 at 1000% strain and 30 at 600% strain with a resistor) without encapsulation. By combining our solder with thermoforming, 3D conformable electronics have been constructed with good electricity stability, easy to process, continuous substitution of electronic components, and circuit recyclability. As examples of emerging applications, an integrated 3D circuit with logic control functions and a stretchable heater is fabricated to show the huge potential of our materials in wearable electronics.

Overall, this study provides an alternative method of developing LM-based self-solder with high conductivity, high stretchability, improved toughness, and excellent weldability that are favorable in the applications of 3D electronics. Despite there are limitations, such as limited resolution (500 μm) due to the solvent evaporate strategy. We believe it is an important step toward reducing the complexity of

microchip interfacing, and thus for scalable fabrication of chip-integrated stretchable circuits and 3D electronics. In addition to the synthesized polymers and LM used in this work, our design concept and fabrication method can be applied to other synthetic or engineered polymers and different liquid metals (Supplementary Fig. 32). We also envision that there will be a lot of opportunities in the design and application of small-molecule modulators to facilitate the assembly and mechanical regulation of polymer composites. Additionally, the new insight into the intermolecular interactions between the supramolecular and the oxide layer of LMP revealed in this research, can largely enrich the knowledge and tools for designing high-performance LMP and multifunctional LM-polymer composites. Even though we highlight the performance of ULPC without an encapsulation layer or chemical modification, these additions can be incorporated for improving the adhesion and electrical stability according to practical applications (Supplementary Fig. 33).

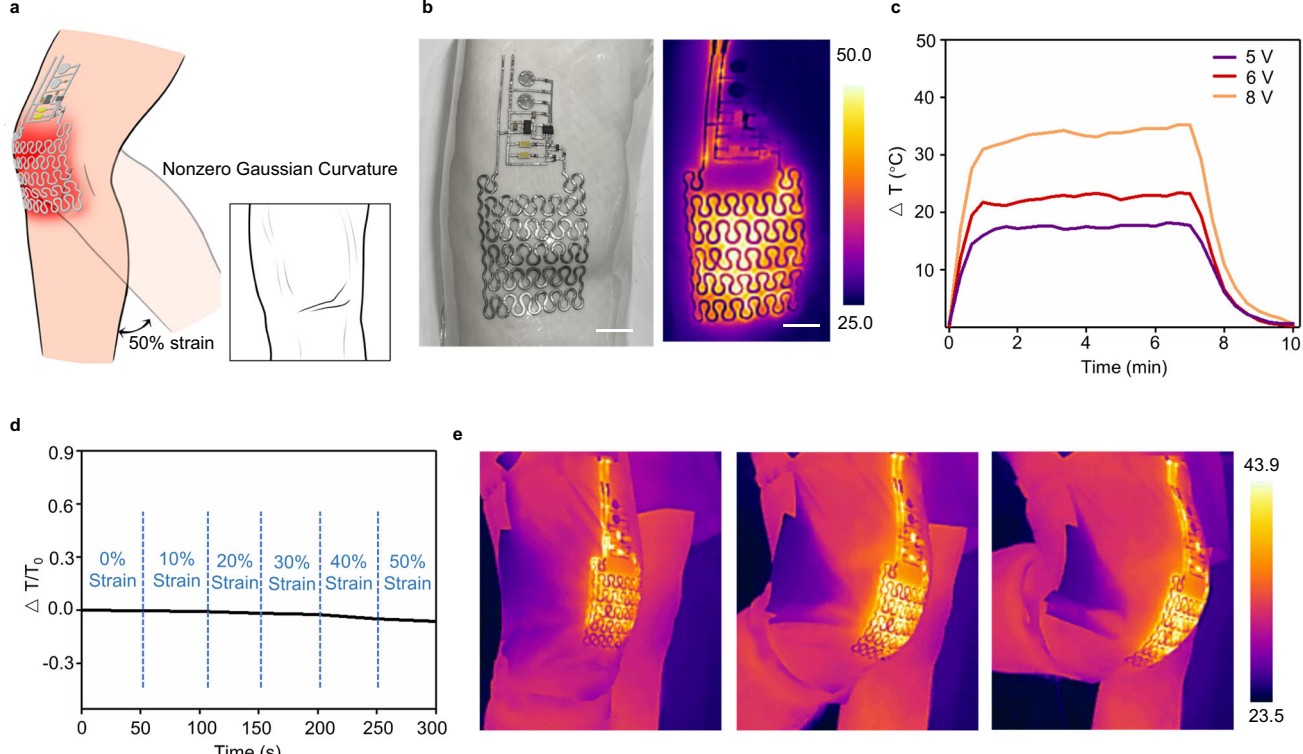

**Fig. 5 | Applications in 3D conformable electronics. a** Overview of the conformable electronic patch for intelligent heating. **b** Optical images of the as-prepared 3D electronic patch, scale bar, 5 mm, the resolution is 500 μm. Each experiment was repeated independently 3 times with similar results. **c** Temperature profiles of the electro-resistive heater under different applied voltages. **d** The temperature change of the heaters under tensile deformations. **e** Infrared camera images showing the reliable heating performance of the electronic patch attached to the knee.

## Methods

### Materials

EGaIn alloy (75.5 wt% Ga and 24.5 wt% In), isophorone diisocyanate (IPDI), poly (propylene glycol) bis (2-aminopropyl ether) (OPG-2NH$_2$) were purchased from Sigma-Aldrich. 2-amino-4-hydroxy-6-methyl-pyrimidine (UPy) was purchased from Aladdin. Poly(dimethylsiloxane) bis(3-aminopropyl) terminated (MW$_{PDMSm}$-2NH$_2$) with different molecular weights were purchased from Gelest. All the chemicals used in the current work were used without further purification.

#### Preparation of linear supramolecular polymer

Polymer with 2000 MWs of UPy-terminated siloxane, 0.08 feeding amounts of UPy monomers (MW$_{PDMS}$2000, UPy0.08) was synthesized by the following steps. The prepared UPy-MW$_{PDMS}$2000-NH$_2$ (460 mg, 0.2 mmol), MW$_{PDMS}$3000-2NH$_2$ (1.73 g, 0.58 mmol), poly (propylene glycol) bis (2-aminopropyl ether) (OPG-2NH$_2$) (230 mg, 0.58 mmol) were dissolved in chloroform (20 mL). In addition, isophorone diisocyanate (IPDI) (277.9 mg, 1.25 mmol) was dissolved in chloroform (5 mL). After that, the above solutions were mixed and stirred in a beaker for 6 h. The precipitate was taken out and washed with acetone three times. Then the product was dried under a vacuum at 70 °C for 24 h.

Other polymers were prepared through similar procedures with corresponding MWs of UPy-terminated siloxane and feeding amounts of UPy monomers.

### Preparation of small-molecule modulator (UPy$_{MC}$)

The prepared UPy-NCO (292.0 mg, 1.00 mmol) was dissolved in chloroform (10 mL), then methyl carbamate (90.1 mg, 1.2 mmol) dissolved in chloroform solution was added dropwise. After reacting at 60 °C under the protection of nitrogen for 6 h, the solvent chloroform was removed, and the solid product was washed 3 times with 30 mL portions of acetone to remove unreacted methyl carbamate. UPy$_{MC}$ was then collected by filtration and dried overnight under a high vacuum at 60 °C (yield 91%).

UPy$_{Gly}$, UPy$_{BI}$ were prepared through similar procedures using corresponding amino terminated monomers.

### The fabrication of ULPC solder

Firstly, 1.5 g polymer was dissolved in 5 mL DCM under magnetic stirring for 30 min. The UPy$_{MC}$-LMP were made by adding 650 mg of EGaIn alloy (75.5% Ga, 24.5% In) and 30 mg UPy$_{MC}$ into 10 mL of DCM and then sonicating at an amplitude of 36 μm (30% setting) for 5 min using a tip sonicator in an ice-water bath. Then, pour the dissolved polymer solution into the UPy$_{MC}$-LMP and continue to sonicate for 1 min to ensure uniform dispersion. The mixture was then poured into glass molds into films by slowly drying at ambient conditions for 1 h for complete drying to obtain ULPC solder.

### Fabrication of circuit

Firstly, the LM-rich surface of ULPC film was attached to the thermal release tape and then patterned by a cutting machine (Sillouette Cameo 4). In step 2, we flipped the circuit pattern ULPC film and thermal release tape together and attached them to the TPU substrate. When the temperature of the TPU substrate was heated up to 100 °C and kept for 1 min, the thermal release tape was released and the circuit pattern ULPC film was mounted to the TPU substrate. In step 3, similar to pick and place technology, we placed surface-mounted devices chips and components on the circuits. In step 4, the circuit was placed on a heating table (120 °C) to adhere electronic components to ULPC-TPU.

### Thermoforming process

The 3D mold used during the thermoforming process was fabricated by a digital light processing 3D printer with heat resistance resin on a

stainless-steel half sphere. Next, a 2D planar TPU film with the necessary electronics was placed on a thermoforming machine and heated up to the softening point of TPU, making it pliable and ready for shaping. Then, the film was stretched over the surface of the 3D mold. Vacuum pressure was applied from beneath the mold, causing the film to conform to the contours of the mold. Finally, the 3D electronics fabrication was completed by detaching it from the 3D mold.

## Mechanical characterization

The samples were cut into dog-bone patterns ($20 \times 4$ mm). The thicknesses of these patterns were averaged from five measurements ($300 \pm 2.80 \, \mu m$). Tensile tests were conducted on AMETEK TCM 100 with a 50 N force sensor, performed on Instron 5566 at a strain rate of 10 mm/min.

## Electrical characterization

The samples were cut into stripes by a cutting machine ($10 \times 1.3$ mm). The thicknesses of these stripes were averaged from five measurements ($300 \pm 2.80 \, \mu m$), and the thicknesses of LM-rich layer were observed by SEM images ($50 \pm 1.25 \, \mu m$). The sheet resistance was first measured using four collinear, equally spaced probes connected to a Keithley 6500 source meter. Initial conductivity of the ULPC was calculated as $\sigma = l/Rwt$, where $l$, $w$, $t$ and $R$ are the length, width, thickness of LM-rich layer and resistance of the ULPC stripes, respectively. For electromechanical characterization, ULPC stripes were fabricated by welding onto TPU substrates. To eliminate the resistance changes at the interfaces between ULPC and external wires, electrical connections were made by adhering copper tape on each end of the ULPC traces, secured using LM. To test the circuits with ULPC–component interfaces, zero-ohm resistors (1/8 W, 0603 package size), LEDs (1/2 W, 2835 package size) were placed between two ULPC traces. The samples were then subjected to uniaxial tensile loading at 10 mm/min.

## Experiments with human subjects

The experiments with human subjects were performed in compliance with all the ethical regulations under a protocol that was reviewed and approved by the JCC College Human Ethics Sub-committee of City University of Hong Kong (jcc2122ay007a). The authors affirm that human research participants provided informed consent for publication of the images in Figure(s) 5e.

## Reporting summary

Further information on research design is available in the Nature Portfolio Reporting Summary linked to this article.

# Data availability

The authors declare that the data supporting the findings of this study are available within the article and its Supplementary Information files. Extra data or source files are available from the corresponding author.

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

## Acknowledgements

This work was supported by the Research Grant Council of Hong Kong with a No. of CityU11307721 (X.Y.), Collaborative Research Fund (CRF) Hong Kong with a No. of C1006-20WF (X.Y.), Shenzhen Basic Research Program with a No. of JCYJ20210324134009024 (X.Y.), Innovation and Technology Fund with a No. of MHP/030/21 (X.Y.).

## Author contributions

X.Y. conceived the idea and supervised the project. L.A., W.L., C.C., P.L., X.W., D.L., and X.L. conducted the experiments. L.A. carried out the synthesis and characterization of the materials, W.L., P.L., and C.C. performed the electrical study and analyzed the data. Z.Y. and W.L. designed the circuit. L.A. and X.Y. wrote the manuscript. All the authors discussed the results and commented on the manuscript.

## Competing interests

The authors declare no competing interests.
