## [Peer review file · Nature Communications]

REVIEWER COMMENTS

Reviewer #1 (Remarks to the Author):

In this work, a LM-based stretchable conductor was fabricated with hierarchical assemblies of LM particles, small-molecule modulators and non-covalently crosslinked polymer matrix, displaying a good weldability and adhesivity towards electronic components. Both the results and experimental characterization are good. But before further consideration on its novelty, the following issues are highly suggested to be resolved.

1. The authors claimed that "Our conductor shows high conductivity ($>40000 \text{ S m}^{-1}$), extreme stretchability ($\sim 1000\%$), and high toughness ($\sim 20 \text{ MJ m}^{-3}$).". Are these results obtained on the composite with specified constituents?
2. A layer covered on the surface of LM particle in Fig. 2b. So, are they the small-molecule modulators like UPy? But it may be very difficult to form a clear layer because they are very small?
3. Why did the LM particles show a continuous form in Fig. 1b and 3a? Need detailed explanation.
4. The effect of LM content on the mechanical properties should be investigated? The sample details like LM content in all figure captions should be given.
5. In Fig. 3i, under a strain of 1000% the resistance of the composite film increased about 10 times. Is it low? A comprehensive comparison is suggested to make like with Nat. Mater. 20, 859–868 (2021).
6. The fabrication process for integrated circuit as shown in Fig. 4a seems cumbersome. Can this composite or its suspension be used as ink for direct writing and 2/3D printing?

Reviewer #2 (Remarks to the Author):

1. General comments:

I am familiar with the stretchable electronics space, and this paper was a pleasure to read.

Heading into the review process, the paper has a very professional feel. However, the authors set themselves up for unnecessary challenges by using the word "weldable" and not backing it up. I read in a non-conventional order - supplementary first - since I was excited to see what details the authors wanted to investigate beyond the flashy public-facing claims in the main text. To my surprise, there was not a deep dive into verification of some sort of true welding in the sense of metal-metal fusing together into less-distinct grains, or something similar using a wider conception of "welding". This was a letdown, since I thought the authors had made a magical solder that could stretch. Unnecessary letdown, though, since the paper was still interesting.

The introduction is well-written, and the problem chosen is timely. Many stretchable electronics fail at the interface, and most liquid-metal-embedded elastomers do indeed simply mix LM into the polymer, ignoring the chemical interaction between LM (or oxide) and polymer. The solution - functionalizing the polymer matrix - is shown to yield impressively robust circuits in the absence of another encapsulation layer. Finally, the authors briefly also show how the circuit components can be removed and re-applied, and also the circuits can be dissolved in a solvent to aid in recyclability.

I think the title would be better if shortened. For example:

"tough electronics enabled by small-molecule modulated interfacial assemblies"

or

"tough circuits enabled by small-molecule modulated interfacial assemblies"

After finishing the main paper, I do think that the supplementary materials go into high-quality science that supports the paper. Very few unnecessary experiments were included, which can be a

drawback for some papers in related fields.

I am curious how this experiment / process would work with a biphasic material [1] such as O-GaIn [2] (oxidized Gallium-Indium), introduced into the matrix, instead of a liquid metal. Maybe it wouldn't mix at all, maybe it would have interesting properties. Just a thought.

[1] Sanchez-Botero, Shah 2022 <https://onlinelibrary.wiley.com/doi/abs/10.1002/adma.202109427>

[2] Wang 2019 <https://onlinelibrary.wiley.com/doi/abs/10.1002/adfm.201907063>

Publishing groundbreaking new work in the stretchable electronics field is difficult, since it is difficult to distinguish and evaluate the differences between such similar papers - there are dozens of stretchable conductors lasting higher than 100% strain, and the practical significance of stretching farther than that is limited in the applications typically listed (soft robotics & wearables, which both only need strains of the 50-100% range maximum). By simply presenting normalized resistance over a single cycle, resistance after cycling to 50% strain, and some application-focused demonstrations, I think this manuscript did not do justice to the technology that is claimed to be introduced.

- How could you more fully quantify the interfacial strength, and compare to other stretchable circuits? Either in shear loading or normal loading. Supplemental materials mentions the stud pull test, and this was used in 3C for LM content 10-40%, as well as 4f for a zero-ohm resistor on the substrate.

- How does the adhesion depend on the semiconductor package type? QFN, TSOP, SOIC... Also, how does the circuit maximum strain and cyclic stability depend on the package type? Could the novel chemistry introduced here be exploited to super-bond to some of these packages?

- How could the chemistry be further modified to maximize adhesion?

- What are the mechanisms for interfacial failure? How do these lead to the massive disparity between maximum strain (600 to 1000% strain vs. cyclic tests only done at 50%)? Is there some sort of Weibull distribution of number of cycles to failure as a function of strain value?

The circuits chosen in the manuscript and also the supplementary videos were not particularly novel or interesting. Put another way, it wasn't clear how the proposed adhesive conductor enabled truly novel/groundbreaking/useful circuits that haven't been demonstrated before (see the many examples in the authors' references, or the ones I'm pasting throughout this review). I encourage the authors to choose an application that requires significant strength under shear - that's where their material should vastly outperform state of the art. While most authors test under tension, the presented ULPC interfaces could be tested under shear. If the interface is as stable as the authors claim in this manuscript, this will be really easy to demonstrate against a control case. The more control cases considered, the better.

Finally, the conclusion section doesn't really say what the limitations are and how to move forward in the future. I would have liked to see some indication of where the authors are going and how other adjacent developments could be brought together with this work to enable some interesting applications or discoveries. The current future works strike an odd combination of vague and specific, leaving me unclear with what the next steps are and why I should be excited.

To summarize my comments in a single sentence: I think the science of the paper is quite promising, but the demonstration of broad applicability & generality is not at the top tier at the moment.

2. Specific comments:

P.5 The Design Principle and Polymer Synthesis

1. The synthesis process (p. 6-7) should be shown in an illustration. Also, the components in Fig. 1c could be more clearly labeled. LM  LMP. Maybe draw box around polymer & UPyMC, or explicitly state in the caption that the color scheme in b exactly matches c. Related, I think the polymer and blue-box in b should both be labeled, not just the polymer. I'm reading this as though you mean to

say that the whole blue region is polymer, and you're showing some cartoonishly large polymers in addition, for clarity.

Then, maybe map that down into the diagrams in d. Also, as a non-chemist, the figure 1d is not clear. I think the three illustrations are supposed to show the same thing, the bottom-left seems to be different than the others.

2. What does the MC stand for in the UPyMC term? I don't see how MC maps to "ester functionalized". Again, maybe this is obvious to chemists, but Nature Communications is read by a much wider audience so please consider your abbreviations carefully. Maybe add an appendix explaining the naming in further detail?

P.8 High Toughness and High Stretchability of ULPC

3. "Due to the lack of interfacial interaction, direct incorporation of LMP into polymer leads to the sharp deterioration of mechanical strength as previously reported" I don't think this is true as a general statement. Please provide an explanation on how your work fits in with what is shown in Kazem 2018 [3] and related work on liquid-metal embedded elastomers [4] [5]. Something weird/interesting is going on with your samples. If possible, I'd even like to see a comparison utilizing one of those techniques to make a "control" Polymer-LM that is as close as possible to some other LMEE you find in the literature (doesn't have to be from the citations listed, obviously - use your judgement).

[3] Kazem 2018 <https://onlinelibrary.wiley.com/doi/abs/10.1002/adma.201706594>

[4] Bartlett 2017 <https://www.pnas.org/content/114/9/2143>

[5] 2017 Review paper <https://onlinelibrary.wiley.com/doi/abs/10.1002/adma.201605985>

P.9 Interface-Dependent Electromechanical Properties of ULPC.

4. Misc. Comments

- Make Micro-CT larger

- Great idea to compare the bulk ULPC with ULPC-TPU substrate. Partially expected result, which is a good thing.

- "In addition to stable electricity"  "In addition to stable electrical resistivity"

- Ashby plot is nice, but where would the following go on this chart: LMEE, biphasic gallium-indium (either with fillers added [6], or natively sourced Ga₂O₃ [1]), or a LM fibre mat [7]

[6] Daalkhaijav 2018 <https://doi.org/10.1002/admt.201700351>

[7] Ma 2021 <https://www.nature.com/articles/s41563-020-00902-3>

- I hope the authors used 4-point method for measuring resistance throughout, since resistance is so low. Use of a 2-terminal technique will introduce excessive error.

- 700% strain without encapsulation is, indeed, impressive, even if on a small sample size.

P.13 Applications in Flexible Electronics

5. What is the minimum feature size? Transfer printing is typically limited to ~500 μm in my experience. Too small of features will lead to poor trace conductivity, poor adhesion, etc. in most stretchable circuit architectures. Could you use a laser-assisted process to improve resolution?

6. I like the thought toward recycle-ability. While it may be true that most people would not want to recycle using DCM, the fact that it can be done at all with your architecture is still interesting. I'd have liked to see more on the recovery yield, the number of times the material can be re-used, etc..

Supplementary Material

I appreciate the supplementary materials. The authors have extensively detailed their experimental systems in a concise manner.

Supplementary Table 1: what method was used to determine Young's modulus? Secant? Tangent along the region that was determined to be the linear region? The stress at 100% strain? (I.e. the "100% modulus" many commercial vendors, including SmoothOn, often publish)

Supplementary Fig. 7: the dashed lines vs. solid are difficult to differentiate, since your datarate was so high. Solutions could be to downsample (not ideal) or make a more sparse dashing, for example.

Supplementary Fig. 10: The droplet-pinning profiles here are illustrative and remarkable. I have seen similar things in other papers, but this presentation is very clear and the difference between the three conditions is stark. In particular, the case where HCl removed the oxide skin graphically shows the lack of pinning/wetting in a very graphic and intuitive manner.

Supplementary Fig. 11: Not sure what is being shown here, since the lighting and color don't seem to be consistent. Can you provide additional images, maybe one is grayscale, another is colored?

Supplementary Fig. 13: How does the R vs. strain curve compare to Pouillet's law? (Quadratic; sometimes mis-called "Ohm's Law").

Supplementary Fig. 14: I'd like to see some more description in the caption or supplementary text to give context to this plot and/or show the main conclusions. Simple, 1 or two sentences - not asking for a whole paragraph per figure. Similar comment about most of the supplementary figures. They're great figures, but the current presentation requires the reader to have significant background in this field or to have fully read the main paper to understand what's going on.

Supplementary Fig. 15: Love the illustration of the "recovery" of the stress-strain curve and hysteresis loop after resting for 2 hrs. It clearly gives a sense of how the material behaves over multiple timescales. I also like the authors' self-control in not calling this self-healing.

Supplementary Fig. 18: "Due to the different solubility of TPU and ULPC in DCM, the ink and the components are separated from the TPU substrate." - Do you mean that the TPU didn't dissolve, while the LM droplets and electronic components did get dislodged (not quite dissolved)? If so, please say something to that effect rather than the vague phrase used in caption.

Supplementary Fig. 19: Dark-blue is difficult to differentiate from the black. Suggest changing one of the colors (light blue?)

Supplementary Fig. 20: Difficult to see bottom error bars on each of the data entries. I know that the error bars are something like 1 SD (also, please specify the exact dispersion metric they represent) and are assumed to be symmetric, but still it's easier to interpret this graphical presentation if there is sufficient contrast between error bars and data bars.

-Typo "Clycles"  "Cycles"

Reviewer #3 (Remarks to the Author):

In this study, the authors developed a tough and weldable conductor using supramolecular polymers and liquid metal particles. The conductor exhibited decent conductivity (maximum 47000 S/m) and

high toughness, which are crucial attributes for the development of soft electrical circuits. Unfortunately, upon careful evaluation, I cannot recommend the publication of this article. The main novelty and selling point of this research lies in the "robust integration with chips." However, all the demonstrations provided in the manuscript do not showcase this feature adequately. Instead, they primarily depict circuit operation under moderate levels of deformation, which do not require the robust integration that is the core focus of this study. The deformation observed in the video clip, particularly the stretching of the LED in the horizontal direction, can easily withstand mechanical deformation and the on-skin demonstration only involves minimal deformation. Furthermore, wireless circuit demonstrations do not require significant mechanical deformation. For the publication in Nature Communications, it is crucial to conduct more appropriate demonstrations to ensure that the main concept of this research is effectively conveyed to both reviewers and readers. Therefore, for this article to be considered for publication, it requires solid demonstrations that showcase the robust integration of the developed conductor with electrical chips. Specifically, it is crucial to provide evidence of the circuit's functionality under harsh strain conditions and repeated deformation, involving multiple chips.

Other concerns are as follows:

1. What is the driving force behind the sintering process between LMP? Are additional activation steps required for electrical activation?
2. The overall quality of the figures is not appropriate for publication in Nature Communications. The schematic illustrations should be redrawn to improve readability. Additionally, the figures need more information. For example, in Figure 2d, each chemical component can be included in the figure.
3. How can the integration of an LM-rich region be made more robust? Since LM exhibits fluidity and instability, what methods can be employed to effectively integrate a chip with a conductor?
4. Is a printed LM conductor mechanically stable? The LM-rich region appears to be susceptible to damage or instability when subjected to rubbing.
5. What level of detail can be achieved in patterning a conductor? Is it possible to achieve high-resolution patterning while maintaining high electrical conductivity for circuit fabrication?

Response to Reviewers' Comments

The Reviewer's comments are in black and revised texts are highlighted.

Reviewer #1

In this work, a LM-based stretchable conductor was fabricated with hierarchical assemblies of LM particles, small-molecule modulators and non-covalently crosslinked polymer matrix, displaying a good weldability and adhesivity towards electronic components. Both the results and experimental characterization are good. But before further consideration on its novelty, the following issues are highly suggested to be resolved.

A: We appreciate the Reviewer for the helpful comments regarding our work. We have revised our work according to the Reviewer's comments. Point-by-point responses are attached below:

Q1. The authors claimed that "Our conductor shows high conductivity ($>40000 \text{ S m}^{-1}$), extreme stretchability ($\sim 1000\%$), and high toughness ($\sim 20 \text{ MJ m}^{-3}$).". Are these results obtained on the composite with specified constituents?

A1: Thanks for the Reviewer's comment. We have optimized the mechanical properties and conductivity of the composite by varying the components of ULPC including base polymer, small-molecule modulator, and liquid metal (LM) content. The reported results of "extreme stretchability ($\sim 1000\%$) and high toughness ($\sim 20 \text{ MJ m}^{-3}$)" were achieved using ULPC prepared from $\text{MW}_{\text{PDMS}2000} \text{UPy } 0.08$, with 2 wt% UPy_{MC} and 30 wt% LM content. We have included this information in the caption of Fig. 2 to clarify the specific constituents used in obtaining these results.

Regarding conductivity, we acknowledge the need to overcome contact resistance and accurately measure small resistances in our conductors. To address this, we employed the 4-point method for measuring resistance throughout the study and recalculated the conductivity accordingly. After making the necessary corrections, the revised conductivity of the ULPC with 30 wt% LM content is reported as $2 \times 10^5 \text{ S m}^{-1}$ in the manuscript. Other related electrical data are also corrected by the 4-point method.

We appreciate your attention to these details and have made the appropriate clarifications in the revised version of the manuscript.

Q2. A layer covered on the surface of LM particle in Fig. 2b. So, are they the small-molecule modulators like UPy? But it may be very difficult to form a clear layer because they are very small?

A2: We are grateful for the Reviewer's comment. In order to clarify the information

about the layer, we have included SEM images along with corresponding element mapping (Ga, O, and C) of UPy_{MC}-LMP in Supplementary Fig. 5. The results clearly show that the C elements are uniformly distributed around the surface of LMP, indicating the presence of a UPy_{MC} layer covering the LM particles. To enhance clarity, we have included explanatory instructions in the caption of Supplementary Fig. 5 to highlight this information.

While it is true that UPy molecules have a small molecular size (fibrillar structures with a diameter of approximately 5 and 14 nm) [1], they possess abundant hydrogen bonds to aggregate into long molecule chains or aggregates [2], which may further allow the assembly on the surface LMP.

[1] E. W. Meijer <https://doi.org/10.1039/C6CC10046E>

[2] E. W. Meijer <https://pubs.rsc.org/en/content/articlelanding/2017/cs/c7cs00564d>

Q3. Why did the LM particles show a continuous form in Fig. 1b and 3a? Need detailed explanation.

A3: We are thankful for the Reviewer's comment. Evaporation-induced sintering of LMP has been proven as a method to generate conductive LM composites [3]. The main reason could be the coalescence of LMPs (i.e. breakage of oxide shell and complete merging) during solvent evaporation, and the addition of UPy_{MC} modulator facilitated the process. In comparison, in the case without UPy_{MC}, there is limited coalescence during evaporation, and the LMPs maintain individual spheres without connections in the LM-rich surface (Supplementary Fig. 9).

To further support our findings, we conducted a study on the settling time of LMP with and without UPy_{MC} modification (Fig. R1). We observed that the LMP precipitated much faster in the dispersion with UPy_{MC} compared to the bare ones. Fig. R1 is added in the Supplementary Information as Supplementary Fig. 12. Therefore, we conclude that the UPy_{MC} modification can improve the interfacial interactions between the LMPs and therefore plays an important role to drive rupture and coalescence of LMPs during solvent evaporation.

Fig. R1 Time-dependent stability of LMP with (right) or without (left) the addition of UPy_{MC} in dichloromethane (DCM) solution for 20 min. LMP with UPy_{MC} shows obvious precipitation within 2 min. While the pure LMP dispersion is still cloudy for

20 min (sonication time: 5 min).

[3] Xiankai Li, 2019, <https://www.nature.com/articles/s41467-019-11466-5>

Q4. The effect of LM content on the mechanical properties should be investigated? The sample details like LM content in all figure captions should be given.

A4: We appreciate the Reviewer's comment. We fixed the content of UPy_{MC} at 2 wt% and compared the mechanical properties with LM content of 0-40 wt% (Fig. R2). From the result, the polymer-UPy_{MC} sample without LM shows higher tensile strength, while the elongation at break has a remarkable decrease due to the high density of hydrogen-bonding crosslinks and the high crystallinity. The addition of 10 wt% LM can improve the stretchability from 480% to 800%. Further increment of the LM content from 10 wt% to 40 wt% would further increase the breaking elongation, while the tensile stress can decrease. The addition of LM would reduce the density of hydrogen-bonding crosslinks and therefore improve the stretchability of the sample. This can be further verified by the XRD results in Fig. 2e.

Fig. R2 is added in the Supplementary Information as Supplementary Fig. 7, and we have added more descriptions to Main Text.

Fig. R2 Comparison of the tensile stress-strain curves for the composites prepared by different LM contents. Polymers were prepared from siloxane oligomers with ~2000-MW and 0.08 amount of UPy monomers, and ULPC was fabricated from the polymer with specific LM content and 2 wt% UPy_{MC}.

Q5. In Fig. 3i, under a strain of 1000% the resistance of the composite film increased about 10 times. Is it low? A comprehensive comparison is suggested to make like with Nat. Mater. 20, 859–868 (2021).

A5: We are grateful for the Reviewer's comment. As suggested by the Reviewer, the resistance change under strain has been corrected with the 4-point method. For a more comprehensive comparison, we have added comparative content to emphasize the adhesion of our materials (Table. R1). Due to space constraints, we have included the table as Supplementary Table 4 in the Supplementary Information.

As from the results presented in the table, the tested resistance change of our composite under strain is not the lowest among other LM-based conductors. This can be attributed to the balance between conductivity and adhesion in the heterostructure conductive layer of our material. However, it is important to note that when integrated with electronic components, our conductor exhibits highly competitive adhesion compared to other stretchable LM-based conductors. This highlights the compatibility of our composite with electronic components.

Table R1 Conductivity, stretchability (with or without chips), relative resistance change

Ref.	Conductor	Stretchability (%)	Stretchability with chip (%)	R/R ₀	Adhesion (N/cm)	Encapsulation
Our work	ULPC	965	700	20	1.6/3.2	No need
[2]	SIS-LM	1200	700	50	<0.1	Need
[3]	bGaIn	1200	500	2	/	No need
[4]	SBS-LM	500	200	500	0.3 _{MPa}	No need
[5]	PSA-LM	950	200	10	1.8	No need
[6]	Fe-LM	300	200	18	/	Need
[7]	Sticker-LM	/	100	2	0.5 _{MPa}	No need
[8]	SWCNT-LM	/	30	1.1	/	No need
[9]	LM SBS mat	2200	/	1	/	/
[10]	Cu-LM	1200	/	40	/	/
[11]	TPU-LM	1000	/	20	<0.1	/
[12]	HRHP-LM	950	/	20	0.35	/
[13]	SEBS-LM	800	/	30	/	/
[14]	PVDF-LM	700	/	400	/	/
[15]	PDMS-LM	500	/	2	/	/
[16]	PU-LM	500	/	1	/	/
[17]	Ni-LM	350	/	7	/	/
[18]	CNF-LM	250	/	100	/	/
[19]	STICK-LM	100	/	1	0.9	/
[20]	PVA-LM	100	/	8	/	/
[21]	TA-LM	70	/	2	/	/

at the maximum strain (R/R_0), adhesion, and encapsulation of various LM-based stretchable conductors.

In applications such as soft robotics and wearables, the maximum strain typically ranges from 50% to 100%. Under this condition, the resistance change observed in our composite film is less than 2 times, which is sufficient to meet the requirements in most practical scenarios.

Q6. The fabrication process for integrated circuit as shown in Fig. 4a seems cumbersome. Can this composite or its suspension be used as ink for direct writing and 2/3D printing?

A6: We appreciate the Reviewer's comment. The evaporation-induced sintering process used to prepare our conductor allows the LMP to form a continuous conductive structure. The amount of LM used in our system is relatively small (10-40 wt%), which may not be ideal for direct writing and 2/3D printing inks that usually require higher LM content.

However, we acknowledge the potential benefits of using our composite or its suspension as ink for these printing techniques. In future research, we will explore and optimize our system to make it more compatible with direct writing and 2/3D printing applications. This may involve modifying the composition, viscosity, and other parameters of the ink to achieve the desired printing properties. We have also discussed it in the conclusion section of the manuscript. We appreciated the comments from the Reviewer, which makes the discussion session rich and meaningful. Furthermore, we have made modifications to Fig. 4a to make it more concise and to better illustrate the process.

Reviewer #2

1. General comments:

I am familiar with the stretchable electronics space, and this paper was a pleasure to read.

Heading into the review process, the paper has a very professional feel. However, the authors set themselves up for unnecessary challenges by using the word "weldable" and not backing it up. I read in a non-conventional order - supplementary first - since I was excited to see what details the authors wanted to investigate beyond the flashy public-facing claims in the main text. To my surprise, there was not a deep dive into verification of some sort of true welding in the sense of metal-metal fusing together into less-distinct grains, or something similar using a wider conception of "welding". This was a letdown, since I thought the authors had made a magical solder that could stretch. Unnecessary letdown, though, since the paper was still interesting.

A: We appreciate the Reviewer's comment and are interested in our work. The intention behind using the term "weldable" was to highlight the strong adhesion between the ULPC conductor and components after thermal treatment. We appreciate your feedback and will take it into consideration for future research and publications.

The introduction is well-written, and the problem chosen is timely. Many stretchable electronics fail at the interface, and most liquid-metal-embedded elastomers do indeed simply mix LM into the polymer, ignoring the chemical interaction between LM (or oxide) and polymer. The solution - functionalizing the polymer matrix - is shown to yield impressively robust circuits in the absence of another encapsulation layer. Finally, the authors briefly also show how the circuit components can be removed and re-applied, and also the circuits can be dissolved in a solvent to aid in recyclability.

I think the title would be better if shortened. For example:

"tough electronics enabled by small-molecule modulated interfacial assemblies"

or "tough circuits enabled by small-molecule modulated interfacial assemblies"

A: We are grateful to the Reviewer for the positive feedback and helpful comment. The title has been changed to "tough electronics enabled by small-molecule modulated interfacial assemblies" as suggested.

After finishing the main paper, I do think that the supplementary materials go into high-quality science that supports the paper. Very few unnecessary experiments were included, which can be a drawback for some papers in related fields.

A: We appreciate the Reviewer for the acknowledgment of our work.

I am curious how this experiment/process would work with a biphasic material [1] such as O-GaIn [2] (oxidized Gallium-Indium), introduced into the matrix, instead of a LM. Maybe it wouldn't mix at all, maybe it would have interesting properties. Just a thought.

[1] Sanchez-Botero, Shah 2022
<https://onlinelibrary.wiley.com/doi/abs/10.1002/adma.202109427>

[2] Wang 2019 <https://onlinelibrary.wiley.com/doi/abs/10.1002/adfm.201907063>

A: We are thankful to the Reviewer for the helpful comment. It is interesting to investigate the O-GaIn in our system. From the result, it does not show apparent difference between LM and O-GaIn (Fig. R3). The resulting composite also shows high conductivity ($2.20 \pm 0.12 \times 10^5$ S/m, stretchability ($\sim 1000\%$), and adhesion (1.80 ± 0.18 MPa) with 30 wt% O-GaIn content.

Fig. R3 a, Optical images of prepared O-GaIn polymer composite. **b,** Strain of the composite to 1000%.

These findings highlight the versatility and potential of our approach in utilizing different materials, such as O-GaIn, to achieve desired properties in stretchable electronics. By demonstrating that O-GaIn can be effectively integrated into the matrix without compromising key performance metrics, you have expanded the range of materials that can be explored for future applications.

Publishing groundbreaking new work in the stretchable electronics field is difficult, since it is difficult to distinguish and evaluate the differences between such similar papers - there are dozens of stretchable conductors lasting higher than 100% strain, and the practical significance of stretching farther than that is limited in the applications typically listed (soft robotics & wearables, which both only need strains of the 50-100% range maximum). By simply presenting normalized resistance over a single cycle, resistance after cycling to 50% strain, and some application-focused demonstrations, I think this manuscript did not do justice to the technology that is claimed to be introduced.

A: We appreciate your valuable feedback and totally agree with the points you have raised regarding the cycling test and the stretchable demonstration. To address these concerns and better showcase the novelty and significance of our work, we have incorporated additional experimental investigating the long-term stability and

durability of our stretchable conductor under cyclic loading, including increasing the number of cycles and strains. Fig. R4a is added in the Main Text as Fig. 3i, and Fig. R4b is added in Supplementary Information as Supplementary Fig. 21.

Fig. R4 a, Resistance stability of chip-integrated ULPC-TPU in 10000 stretch-release cycles at 50% strain, and **b**, 2500 stretch-release cycles at 100% strain.

Furthermore, we have expanded on the practical applications of our stretchable conductor. In particular, we have focused on thermoforming 3D circuits to demonstrate the unique ability of our material to resist the enormous shear forces generated during this process. We believe that this application showcases the potential of our technology beyond the typical soft robotics and wearables, providing a broader perspective on its practical significance.

We hope that these additions and enhancements better highlight the novelty and significance of our work. Thank you for your constructive feedback, which has greatly contributed to the improvement of our manuscript.

- How could you more fully quantify the interfacial strength, and compare to other stretchable circuits? Either in shear loading or normal loading. Supplemental materials mentions the stud pull test, and this was used in 3C for LM content 10-40%, as well as 4f for a zero-ohm resistor on the substrate.

A: Thank you for your inquiry regarding the quantification of interfacial strength. By performing the stud pull test, we were able to measure the force required to separate the bonded components. In order to compare our conductor with other LM-based stretchable conductors, we also performed a 90° peel-off test to measure the maximum shear force required to cause the interface between ULPC and the metal substrate (Cu film) to fail. From the result, the adhesion of the two sides of ULPC was found to be different due to the different composition. Notably, when compared with other LM-based stretchable conductors, our ULPC demonstrated competitive performance, as indicated in Table R1. This suggests that our chip-integrated conductor has the ability to withstand shear force. Fig. R5 is added in Supplementary Information as

Supplementary Fig. 14.

Fig. R5 The tensile curves for the 90° peel-off test. The pull-off strength was measured according to the forces and the contact area of the Cu films and ULPC with 30 wt% LM content. The mechanical test machine with a constant speed of 50 mm/min.

- How does the adhesion depend on the semiconductor package type? QFN, TSOP, SOIC... Also, how does the circuit maximum strain and cyclic stability depend on the package type?

A: Thanks for pointing this out, it is important to study the effect of different package types on stretchable circuits. In our study, we have incorporated various package components such as SOT-23, SOIC-8, 0805 or 1206 components, and 5730 SMD LED. The adhesion between ULPC and electronics components relies on their contact area. The larger the contact area, the stronger the adhesion is likely to be. It turns out that our conductor is suitable for achieving reliable adhesion and connection with these package types. However, it is important to note that for a typical QFN package chip, with a pin width of 300 μm and a gap of 500 μm , our circuit may not be suitable due to fabrication resolution.

-Could the novel chemistry introduced here be exploited to super-bond to some of these packages?

- How could the chemistry be further modified to maximize adhesion?

A: We appreciate the Reviewer for the comment. One possible way to improve the adhesion is to perform suitable surface treatment of the component and the stretchable conductor. This can be achieved through surface functionalization techniques, such as plasma treatment or chemical modification, to create a more reactive surface that promotes stronger bonding with the binding polymer. As shown in Fig. R6, the peel-off strength shows visible improvement after surface treatment.

Fig. R6 Comparison of the maximum peel-off strength by different surface treatment approaches. The plasma is conducted with 40 W/ 3 min, the chemical modification of hydroxyl was put components in ethanol ultrasound for 5 min, then place in a mixture of sodium hydroxide and ethanol (1:1) ultrasound for 5 min, and dry them with nitrogen.

- What are the mechanisms for interfacial failure? How do these lead to the massive disparity between maximum strain (600 to 1000% strain vs. cyclic tests only done at 50%)? Is there some sort of Weibull distribution of number of cycles to failure as a function of strain value?

A: We are grateful to the Reviewer for the insightful comment. The significant disparity between the maximum strain (ranging from 600% to 1000%) and the cyclic tests performed at only 50% strain can be attributed to the different modes of loading and the associated stress levels. During the maximum strain tests, the materials experience a one-time extreme deformation. In contrast, cyclic tests involve repeated loading and unloading cycles, which induce fatigue and cumulative damage over time. The lower strain level in cyclic tests helps to ensure the longevity and reliability of the circuit under repeated stress.

To further investigate the mechanisms for interfacial failure, we also increase the cycles of the cyclic test from 1000 to 10000 and added 100% strain cycle test for a more comprehensive understanding of material properties (Fig. R4). Under 50% strain, in the first 5000 cycles, the R_0 value (resistance at 0% strain) increases slightly. However, this increase is only from the initial value of $\approx 0.7 \Omega$ to a final value of $\approx 1.1 \Omega$ and then stabilizes. Under 100% strain, the resistance increases slightly in the first 1000 cycles. Then, gradually increase from 1.5Ω to 2.1Ω , and fail in ≈ 2500 cycles, due to interfacial failure.

Fig. R4 a, Resistance stability of chip-integrated ULPC-TPU in 10000 stretch-release cycles at 50% strain, and **b**, 2500 stretch-release cycles at 100% strain.

Regarding the Weibull distribution of the number of cycles to failure as a function of strain value, this is an interesting concept that can provide insights into the reliability and lifetime prediction of the circuits. While we have not specifically investigated the Weibull distribution in this study, it is a potential avenue for future research to analyze the relationship between strain values and the number of cycles to failure.

The circuits chosen in the manuscript and also the supplementary videos were not particularly novel or interesting. Put another way, it wasn't clear how the proposed adhesive conductor enabled truly novel/groundbreaking/useful circuits that haven't been demonstrated before (see the many examples in the authors' references, or the ones I'm pasting throughout this review). I encourage the authors to choose an application that requires significant strength under shear - that's where their material should vastly outperform state of the art. While most authors test under tension, the presented ULPC interfaces could be tested under shear. If the interface is as stable as the authors claim in this manuscript, this will be really easy to demonstrate against a control case. The more control cases considered, the better.

A: Thank you for your valuable feedback. In response, we would like to introduce a novel application named 3D conformal electronics for promoting human joint health through vacuum thermoforming. While many existing planar circuits are stretchable, they often struggle to meet the requirements of complex human body surfaces, such as knee joints, which exhibit non-zero Gaussian surfaces and a wide range of motion.

Although 3D printing or kirigami techniques can draw circuits on complex surfaces, the challenge lies in placing electronic components on curved surfaces. Commercial pick-and-place technology, commonly used for fabricating planar circuits, is not suitable for this purpose. To address these challenges, we have developed a simple and efficient method for fabricating functional circuits on a 2D planar thermoplastic elastomer (TPU) substrate. We then bond all electronic components to the conductive

path of the circuit.

Next, we employ vacuum thermoforming to precisely conform the planar circuit to a 3D mold. After cooling, a stretchable 3D circuit that perfectly fits complex curved surfaces is obtained. One of the main challenges we encountered was the large shear force exerted on electronic components during the vacuum thermoforming process, potentially resulting in circuit failure. However, our materials demonstrate strong adhesion to electronic components, surpassing commercial silver paste in performance.

By addressing these challenges, our approach enables the creation of stretchable 3D circuits that can conform to complex curved surfaces like knee joints. We believe that our new application can fully demonstrate the advantages of our material, strong adhesive, high conductive, and high stretchability. We also revise the whole of Fig. 4 and Fig. 5, adding more schematics to illustrate our new applications and highlight the importance of our materials in the practical application.

Finally, the conclusion section doesn't really say what the limitations are and how to move forward in the future. I would have liked to see some indication of where the authors are going and how other adjacent developments could be brought together with this work to enable some interesting applications or discoveries. The current future works strike an odd combination of vague and specific, leaving me unclear with what the next steps are and why I should be excited.

A: We appreciate the Reviewer's suggestion regarding the limitations and future directions of our work. We acknowledge that the current conclusion section may not adequately address these aspects.

In terms of limitations, we recognize that the current resolution capabilities of cutting and thermal transfer printing technologies used in our circuit fabrication process are limited to 300 μm and 500 μm , respectively. This restricts the application of our approach when it comes to smaller and more complex circuits. We have highlighted this limitation in the revised conclusion section.

Regarding future directions, we agree that the description of our future works may have appeared somewhat vague and specific. Moving forward, we aim to explore opportunities for integrating our findings with other adjacent developments to enable exciting applications or discoveries. One potential avenue we are considering is adjusting the solvent of ULPC ink, which could potentially enhance the precision of ULPC conductor traces when using screen printing or 2D/3D printing technology.

Additionally, we have expanded on the practical applications of our stretchable conductor. In particular, we have focused on thermoforming 3D circuits to demonstrate the unique ability of our material to resist the enormous shear forces generated during this process. We believe that this application showcases the potential of our technology

beyond the typical soft robotics and wearables, providing a broader perspective on its practical significance. We have added a more specific description in the conclusion section.

Once again, we sincerely appreciate the Reviewer's feedback. We hope that these additions and enhancements better highlight the novelty and significance of our work. Your input has been incredibly valuable in improving the manuscript, and we thank you for your constructive comments.

To summarize my comments in a single sentence: I think the science of the paper is quite promising, but the demonstration of broad applicability & generality is not at the top tier at the moment.

A: We are thankful to the Reviewer for the positive evaluation of our work and constructive suggestions for improving our job. We have revised our work according to the Reviewer's comments. Point-by-point responses are attached below:

2. Specific comments:

Q1. P.5 The Design Principle and Polymer Synthesis

The synthesis process (p. 6-7) should be shown in an illustration. Also, the components in Fig. 1c could be more clearly labeled. LM  LMP. Maybe draw box around polymer & UPyMC, or explicitly state in the caption that the color scheme in b exactly matches c. Related, I think the polymer and blue-box in b should both be labeled, not just the polymer. I'm reading this as though you mean to say that the whole blue region is polymer, and you're showing some cartoonishly large polymers in addition, for clarity.

Then, maybe map that down into the diagrams in d. Also, as a non-chemist, the figure 1d is not clear. I think the three illustrations are supposed to show the same thing, the bottom-left seems to be different than the others.

A1: We appreciate the Reviewer for the careful and helpful comments. We have taken note of your comment and redrawn Fig.1 to ensure clarity and consistency.

Fig. R7 Design principle of the LM-based composite conductor.

Q2. What does the MC stand for in the UPyMC term? I don't see how MC maps to "ester functionalized". Again, maybe this is obvious to chemists, but Nature Communications is read by a much wider audience so please consider your abbreviations carefully. Maybe add an appendix explaining the naming in further detail?

A2: We are grateful to the Reviewer for the helpful comments. In this manuscript, "MC" stands for "Methyl Carbamate," which is the full name of the monomer that reacts with UPy-NCO. We understand that abbreviations can sometimes be unclear to a wider audience, and we appreciate your suggestion to provide further clarification. To address this concern, we add an instruction in Supplementary Fig. 3 that explains the abbreviations and naming conventions used in the manuscript.

Q3. P.8 High Toughness and High Stretchability of ULPC

"Due to the lack of interfacial interaction, direct incorporation of LMP into polymer leads to the sharp deterioration of mechanical strength as previously reported" I don't think this is true as a general statement. Please provide an explanation on how your work fits in with what is shown in Kazem 2018 [3] and related work on liquid-metal embedded elastomers [4] [5]. Something weird/interesting is going on with your samples. If possible, I'd even like to see a comparison utilizing one of those techniques to make a "control" Polymer-LM that is as close as possible to some other LMEE you find in the literature (doesn't have to be from the citations listed, obviously - use your judgement).

[3] Kazem 2018 <https://onlinelibrary.wiley.com/doi/abs/10.1002/adma.201706594>

[4] Bartlett 2017 <https://www.pnas.org/content/114/9/2143>

[5] 2017 Review paper
<https://onlinelibrary.wiley.com/doi/abs/10.1002/adma.201605985>

A3: We appreciate the Reviewer for the helpful comment. Thank you for bringing up the relevant literature on LM composites and for providing specific references.

After examining the literature Reviewer mentioned, we agree that the mechanical properties of LM composites can vary depending on various factors, including the size of the LMP and the fabrication process. In our study, we observed a deterioration of mechanical strength after the direct incorporation of LMP into the polymer matrix. This observation is consistent with some previous reports where the addition of larger-sized LMP led to a decrease in mechanical properties [4-6]. In our case, the LMP particles were ultrasonicated for a relatively short duration. During the solvent volatilization process, the LMP particles tend to aggregate and form large connected networks, which can weaken the mechanical strength of the composite.

To further investigate the effects of LMP and fabrication processes on mechanical properties, the effect of LM content on mechanical properties was also investigated.

From the result, an increase in increment of the LM content from 10 wt% to 40 wt% would decrease the tensile stress of ULPC from 3.2 MPa to 2.7 MPa, while the tensile strain can increase.

To avoid misleading, we changed the sentence in the main text to “It should be noted that the direct incorporation of high mass ratio of LMP (30%) into polymer leads to the sharp deterioration of mechanical strength due to the aggregation of LMP as previously reported”.

Fig. R2 Comparison of the tensile stress-strain curves for the composites prepared by different LM content. Polymers were prepared from siloxane oligomers with ~2000-MW and 0.08 amount of UPy monomers, and ULPC was fabricated from the polymer with specific LM content and 2 wt% UPy_{MC}.

[4] Xin 2021 <https://onlinelibrary.wiley.com/doi/full/10.1002/admt.202000852>

[5] Ma 2021 <https://www.nature.com/articles/s41563-020-00902-3>

[6] Chen 2022 <https://www.nature.com/articles/s41467-022-28901-9>

Q4. P.9 Interface-Dependent Electromechanical Properties of ULPC.

- Make Micro-CT larger)

A: We are thankful to the Reviewer for the helpful comment. The sample size is customized according to the instrument. We have increased the size of the micro-CT image in the manuscript to provide readers with a better view of the structure and enhance the clarity of the image.

-Great idea to compare the bulk ULPC with ULPC-TPU substrate. Partially expected result, which is a good thing.

A: We appreciate the Reviewer for the acknowledgment of our work.

"In addition to stable electricity"  "In addition to stable electrical resistivity"

A: We are grateful to the Reviewer for the valuable suggestion. We have revised the sentence as you recommended, changing "stable electricity" to "stable electrical resistivity" to accurately reflect the intended meaning.

Ashby plot is nice, but where would the following go on this chart: LMEE, biphasic

gallium-indium (either with fillers added [6], or natively sourced Ga₂O₃ [1]), or a LM fibre mat [7]

[6] Daalkhaijav 2018 <https://doi.org/10.1002/admt.201700351>

[7] Ma 2021 <https://www.nature.com/articles/s41563-020-00902-3>)

A: We appreciate the Reviewer for the helpful comments. For a more comprehensive comparison, we have added comparative content to emphasize the adhesion of our materials (Table. R1). The suggested references have been added to the table for comparison. Due to space constraints, we have included the table as Supplementary Table 4 in the Supplementary Information. As from the results presented in the table, our conductor exhibits highly competitive adhesion compared to other stretchable LM-based conductors. This highlights the compatibility of our composite with electronic components.

- I hope the authors used 4-point method for measuring resistance throughout, since resistance is so low. Use of a 2-terminal technique will introduce excessive error.

A: We are grateful to the Reviewer for the helpful suggestion. We used the 4-point method for measuring resistance throughout and recalculating conductivity. The initial resistance for ULPC with the 4-point method (4PP) is 0.71 Ω , while the value obtained with the 2-point method (2PP) is 1.58 Ω (2.2 times larger), owing to experimental errors introduced by the connection between the measurement device and the sample. Besides, we also compare the relative resistance change of ULPC over strains with 4PP, 2PP and Pouillett's Law. The result shows a higher change in resistance for the 4PP curve, R/R_0 increased to 21.3 when $\epsilon=1000\%$ for 4PP method, and 11.7 for 2PP method. This is expected since the 4PP method has a smaller initial resistance. So, we recalculate conductivity with the initial resistance value obtained with the 4PP method, the related data was revised in the Main Text.

Fig. R8 Electromechanical behavior of ULPC. Resistance responses of the samples under a tensile strain up to $\epsilon=1000\%$ with 2PP and 4PP measurement methods (left). Relative change in resistance R/R_0 of ULPC under a tensile strain up to $\epsilon = 1000\%$ with 2PP, 4PP measurement methods and Pouillett's law (right).

- 700% strain without encapsulation is, indeed, impressive, even if on a small sample

size.

A: We are thankful to the Reviewer for the acknowledgment of our work.

Q5. P.13 Applications in Flexible Electronics

What is the minimum feature size? Transfer printing is typically limited to $\sim 500\ \mu\text{m}$ in my experience. Too small of features will lead to poor trace conductivity, poor adhesion, etc. in most stretchable circuit architectures. Could you use a laser-assisted process to improve resolution?

A5: We appreciate the Reviewer for the comment. We have added conductivity data according to the line width in the Supplementary Information. As line width decreased, conductivity decreased likely due to the diminishing of conductive pathways. However, ULPC lines showed decent conductivity over $1 \times 10^5\ \text{S/m}$ at $500\ \mu\text{m}$ width. To achieve reliable electrical conductivity, it is recommended to use ULPC with a width of over $500\ \mu\text{m}$. Fig. R9 is added in Supplementary Information as Supplementary Fig. 27.

Fig. R9 ULPC with different line widths. a, Optical image of ULPC lines with different line widths. b, Conductivity of ULPC line with different line widths.

Regarding the use of a laser-assisted process to improve resolution, we appreciate your suggestion. We have explored the possibility of laser cutting in our system and some preliminary results are shown in Fig. R10. We found that the high temperature generated during the laser-cutting process caused the composite material to melt, leading to reconnected or fracture of the composite and affecting the resolution.

Fig. R10 Optical images of ULPC after laser-assisted cutting

Q6. I like the thought toward recycle-ability. While it may be true that most people would not want to recycle using DCM, the fact that it can be done at all with your architecture is still interesting. I'd have liked to see more on the recovery yield, the number of times the material can be re-used, etc.

A6: We appreciate the reviewer for the comment. We used DCM because of its good solubility for both polymers and UPy_{MC} modulators. To address the concern about the toxicity and corrosion of DCM, we have explored the possibility of recycling using ethanol, which offers better safety and environmental compatibility (Fig. R11). We have added recovery yield and conductivity data according to the number of times the materials were re-used (Fig. R12).

Fig. R11 is added in Supplementary Information as Supplementary Fig. 25. Fig R12 is added in the Main Text as Fig. 4g.

Fig. R11 Recycling process of an integrated 3D circuit. The circuit is placed into an ethanol solution, the ULPC conductors could be easily dissolved and the TPU didn't dissolve. In this way, the ink and the components are separated from the TPU substrate. Alternatively, after removing the components, the concentrated ink can be dissolved in dichloromethane (DCM) and placed to sonicate for 1 min to ensure uniform dispersion and the 3D TPU film can be dissolved in dimethylsulfoxide (DMSO). Then poured into glass molds to re-obtain ULPC and TPU film separately.

Fig. R12 The measured conductivity and yield (mass after cycle relative to initial mass, the mass measured after drying at 60°C for 24 h) of the ULPC over repeated recycling process.

Supplementary Material

I appreciate the supplementary materials. The authors have extensively detailed their experimental systems in a concise manner.

A: Thanks for your positive evaluation of our work.

Q7. Supplementary Table 1: what method was used to determine Young's modulus? Secant? Tangent along the region that was determined to be the linear region? The stress at 100% strain? (I.e. the "100% modulus" many commercial vendors, including SmoothOn, often publish)

A7: We appreciate the Reviewer for the comment. We calculate the Young's modulus by finding the slope of the linear region of the stress-strain graph. The instrument has been added in the caption of Supplementary Table.

Q8. Supplementary Fig. 7: the dashed lines vs. solid are difficult to differentiate, since your datarate was so high. Solutions could be to downsample (not ideal) or make a more sparse dashing, for example.

A8: We are grateful to the Reviewer for the suggestion. We have made the dashing sparser in the figure to improve visibility and differentiation between the lines.

Q9. Supplementary Fig. 10: The droplet-pinning profiles here are illustrative and remarkable. I have seen similar things in other papers, but this presentation is very clear and the difference between the three conditions is stark. In particular, the case where HCl removed the oxide skin graphically shows the lack of pinning/wetting in a very graphic and intuitive manner.

A9: We appreciate the Reviewer for the acknowledgment of our experiment.

Q10. Supplementary Fig. 11: Not sure what is being shown here, since the lighting and

color don't seem to be consistent. Can you provide additional images, maybe one is grayscale, another is colored?

A10: We are thankful to the Reviewer for bringing up this concern and we apologize for any confusion caused. In the figure, we aimed to visually represent the competition between conductivity and adhesion on the surface of ULPC with varying content of LM. In order to avoid any potential confusion or misinterpretation, we have decided to remove the data.

Q11. Supplementary Fig. 13: How does the R vs. strain curve compare to Pouillet's law? (Quadratic; sometimes mis-called "Ohm's Law").

A11: We appreciate the Reviewer for the suggestion. We have added information about Pouillet's law and resistance and strain to Supplementary Fig. 16.

Fig. R13 Comparison of relative resistance change over strains with or without adding UPy_{MC} modulators (left). Normalized change in resistance as a function of strain (gradient of blue solid lines) along with the theoretical prediction using Pouillet's law for an incompressible elastomer with constant volumetric resistivity (dashed black line) (right). The resistance changes at small strain (0~100%) agree well with Pouillet's law.

Q12. Supplementary Fig. 14: I'd like to see some more description in the caption or supplementary text to give context to this plot and/or show the main conclusions. Simple, 1 or two sentences - not asking for a whole paragraph per figure. Similar comment about most of the supplementary figures. They're great figures, but the current presentation requires the reader to have significant background in this field or to have fully read the main paper to understand what's going on.

A12: We are grateful to the Reviewer for the suggestion. We have added more descriptions to the caption of Supplementary Fig. 17 to provide readers with a clearer understanding of the plot and its main conclusions.

Q13. Supplementary Fig. 15: Love the illustration of the "recovery" of the stress-strain curve and hysteresis loop after resting for 2 hrs. It clearly gives a sense of how the material behaves over multiple timescales. I also like the authors' self-control in not calling this self-healing.

A13: We appreciate the Reviewer for the positive evaluation of our experiment.

Q14. Supplementary Fig. 18: "Due to the different solubility of TPU and ULPC in DCM, the ink and the components are separated from the TPU substrate." - Do you mean that the TPU didn't dissolve, while the LM droplets and electronic components did get dislodged (not quite dissolved)? If so, please say something to that effect rather than the vague phrase used in caption.

A14: We are thankful to the Reviewer for the suggestion. We have added more descriptions to the caption of Supplementary Fig. 25 as follows: The circuit is placed into an ethanol solution, the ULPC conductors could be easily dissolved and the TPU didn't dissolve. In this way, the ink and the components are separated from the TPU substrate. Alternatively, after removing the components, the concentrated ink can be dissolved in dichloromethane (DCM) and placed to sonicate for 1 min to ensure uniform dispersion, and the 3D TPU film can be dissolved in dimethylsulfoxide (DMSO). Then poured into glass molds to re-obtain ULPC and TPU film separately.

Q15. Supplementary Fig. 19: Dark-blue is difficult to differentiate from the black. Suggest changing one of the colors (light blue?)

A15: We appreciate the Reviewer for the suggestion. The color has been changed in Supplementary Fig. 23.

Q16. Supplementary Fig. 20: Difficult to see bottom error bars on each of the data entries. I know that the error bars are something like 1 SD (also, please specify the exact dispersion metric they represent) and are assumed to be symmetric, but still it's easier to interpret this graphical presentation if there is sufficient contrast between error bars and data bars.

-Typo "Clycles"  "Cycles"

A16: We appreciate the Reviewer for the suggestion. The error bars and the typo have been revised accordingly.

Reviewer #3

In this study, the authors developed a tough and weldable conductor using supramolecular polymers and liquid metal particles. The conductor exhibited decent conductivity (maximum 47000 S/m) and high toughness, which are crucial attributes for the development of soft electrical circuits. Unfortunately, upon careful evaluation, I cannot recommend the publication of this article.

The main novelty and selling point of this research lies in the "robust integration with chips." However, all the demonstrations provided in the manuscript do not showcase this feature adequately. Instead, they primarily depict circuit operation under moderate levels of deformation, which do not require the robust integration that is the core focus of this study. The deformation observed in the video clip, particularly the stretching of the LED in the horizontal direction, can easily withstand mechanical deformation and the on-skin demonstration only involves minimal deformation. Furthermore, wireless circuit demonstrations do not require significant mechanical deformation. For the publication in Nature Communications, it is crucial to conduct more appropriate demonstrations to ensure that the main concept of this research is effectively conveyed to both reviewers and readers.

Therefore, for this article to be considered for publication, it requires solid demonstrations that showcase the robust integration of the developed conductor with electrical chips. Specifically, it is crucial to provide evidence of the circuit's functionality under harsh strain conditions and repeated deformation, involving multiple chips.

A: We appreciate the Reviewer for the helpful comments regarding our work. We acknowledge the Reviewer's comment regarding the need for more appropriate demonstrations to showcase the robust integration of our conductor with chips, which is a key novelty of our research. To address this concern, we have made significant revisions to the manuscript to include more explicit and compelling demonstrations of 3D conformal electronics for promoting human joint health through vacuum thermoforming. The strong adhesion to electronic components allows our conductor to maintain conductivity under high temperatures and stretching, without electrical failure during the thermoforming process. Our revised approach now showcases the creation of stretchable 3D circuits that effectively conform to complex curved surfaces such as knee joints. We believe that our new application can fully demonstrate the robust integration with chips of our material.

We hope that these revisions effectively address your concerns and that the updated manuscript now effectively conveys the main concept of our research. We appreciate your continued consideration of our work for publication in Nature Communications.

We have revised our work according to the Reviewer's comments. Point-by-point responses are attached below:

Q1. What is the driving force behind the sintering process between LMP? Are additional activation steps required for electrical activation?

A1: We are grateful to the Reviewer for the question. The main reason could be the coalescence of LMPs (i.e. breakage of oxide shell and complete merging) during solvent evaporation, and the addition of UPy_{MC} modulator facilitated the process. In comparison, in the case without UPy_{MC}, there is limited coalescence during evaporation, and the LMPs maintain individual spheres without connections in the LM-rich surface (Supplementary Fig. 9).

To further support our findings, we conducted a study on the settling time of LMP with and without UPy_{MC} modification (Fig. R1). We observed that the LMP precipitated much faster in the dispersion with UPy_{MC} compared to the bare ones. Fig. R1 is added in the Supplementary Information as Supplementary Fig. 12. Therefore, we conclude that the UPy_{MC} modification can improve the interfacial interactions between the LMPs and therefore plays an important role to drive rupture and coalescence of LMPs during solvent evaporation.

Fig. R1 Time-dependent stability of LMP with (right) or without (left) the addition of UPy_{MC} in dichloromethane (DCM) solution for 20 min. LMP with UPy_{MC} shows obvious precipitation within 2 min. While the pure LMP dispersion is still cloudy for 20 min (sonication time: 5 min).

With the compact assembly of LMP, peeling films of ULPC from the substrate can generate sufficient stress to percolate particles [7-8]. Therefore, we don't need additional activation steps for electrical activation. In order to facilitate the reader's understanding, we have drawn a schematic diagram to show the process (Fig. R14) and add instructions in the Main Text.

Fig. R14 is added in the Supplementary Information as Supplementary Fig. 13.

Fig. R14 Schematic illustration of the peeling activation process.

[7] Gun-Hee Lee, 2023, <https://www.nature.com/articles/s41467-023-39928-x>

[8] Lixue Tang, 2018, <https://www.sciencedirect.com/science/article/pii/S2589004218300683?via%3Dihub>

Q2. The overall quality of the figures is not appropriate for publication in Nature Communications. The schematic illustrations should be redrawn to improve readability. Additionally, the figures need more information. For example, in Figure 2d, each chemical component can be included in the figure.

A2: We appreciate the Reviewer's suggestion to improve the overall quality of the figures. We have taken your advice and redrawn the figures to enhance readability and clarity. We also revise the whole of Fig. 4 and Fig. 5, adding more schematics to illustrate our new applications and highlight the importance of our materials in the practical application. We believe that these changes have significantly improved the overall quality of the figures and will effectively convey the necessary information to the readers. Thank you for bringing this to our attention.

Q3. How can the integration of an LM-rich region be made more robust? Since LM exhibits fluidity and instability, what methods can be employed to effectively integrate a chip with a conductor?

A3: We are grateful to the Reviewer for the question. We acknowledge that LM exhibits fluidity and instability in most LM composites. However, in our system, we have employed small molecule modulators to improve the interaction between the LMP and the synthesized polymer, resulting in the formation of a heterogeneously structured LM-rich layer on the surface of ULPC. The surrounding high-toughness polymer compartments can tightly retain LM to avoid leaking. As a result, the LM-rich layer is mechanically and electrically stable even after the welding and thermoforming process.

The rich hydrogen bonding within our conductor enables effectively integrate with the electronic components through thermal processing. The stable adhesion is achieved through molecular bond exchange and recombination when the ULPC conductor comes into contact with electrical components. To further support the electrical stability, we have conducted cyclic tensile loading-unloading tests on the chip-integrated ULPC under 50% strain for 10000 cycles. Additionally, we have performed a 90° peel-off test to measure the adhesion. Comparing our ULPC with other LM-based stretchable

conductors, we find it to be competitive (Table R1). These tests demonstrate the robustness of the integration and validate the electrical stability of the conductor.

Fig. R4 Resistance stability of chip-integrated ULPC-TPU in 10000 stretch-release cycles at 50% strain,

Fig. R5 The tensile curves for the 90° peel-off test. The pull-off strength was measured according to the forces and the contact area of the Cu films and ULPC with 30 wt% LM content. The mechanical test machine with a constant speed of 50 mm/min.

Q4. Is a printed LM conductor mechanically stable? The LM-rich region appears to be susceptible to damage or instability when subjected to rubbing.

A4: We appreciate the Reviewer for the question. In our study, we have specifically designed a heterogeneously structured LM-rich layer that exhibits exceptional mechanical stability. To demonstrate this, we conducted an experiment as illustrated in Figure 3d. We used Scotch tape to repeatedly peel off from the conductive interface of the ULPC. During this process, we observed that the ULPC conductor exhibited only marginal changes in resistance. This demonstrates that the LM-rich layer is highly stable and resistant to damage even under repeated rubbing. Additionally, to quantitatively assess the adhesion of our conductor, we performed a 90° peel-off test with a metal film. The results of this test demonstrate the advantages of our LM-based conductor compared to other existing LM-based conductors (Table R1).

Q5. What level of detail can be achieved in patterning a conductor? Is it possible to achieve high-resolution patterning while maintaining high electrical conductivity for circuit fabrication?

A5: We appreciate the Reviewer for the comment. We have added conductivity data according to the line width in the Supplementary Information. As line width decreased, conductivity decreased likely due to the diminishing of conductive pathways. However, ULPC lines showed decent conductivity over 1×10^5 S/m in 500 μm . To achieve reliable electrical conductivity, it is recommended to use ULPC with a width of over 500 μm . Fig. R9 is added in Supplementary Information as Supplementary Fig. 27.

Fig. R9 ULPC with different line widths. a, Optical image of ULPC lines with different line widths. b, Conductivity of ULPC line with different line widths.

REVIEWER COMMENTS

Reviewer #1 (Remarks to the Author):

The authors have made detailed response to the reviewers' comments. And the manuscript has been improved. I have no further questions.

Reviewer #2 (Remarks to the Author):

Overall, the reviewers have greatly improved the manuscript, and addressed the vast majority of my comments. I do not have major concerns with the manuscript, but will still provide some comments below in the spirit of constructive criticism. This reviewer will look forward to reading this manuscript whenever/wherever it ends up being published, and replicating the results.

1. The "Our work" row in Table R1 - is the adhesion column supposed to have two values? It wasn't clear why two were listed for "Our work" but only one listed for the others.

2. The adhesion experiments are still unclear to me. For example, the stud pull method was mentioned in the SI but I can't figure out which figures used a 90-degree pull test and which used the stud pull test. In particular, Fig. 3 C is driving me crazy since I can't figure out which of the presented techniques was used. I would encourage the authors to double-check the revised manuscript for similar confusions - make sure each figure's technique is clearly mentioned and mention whether it's described in supplementary materials vs. main manuscript's "materials & methods" section. I would further encourage the authors to use clear terminology to reinforce the methods used. For instance, calling Fig. S23 "Peel off strength of zero-ohm resistors" seems like a mistake to this reviewer. I would assume Fig. S23 used stud pull method. For terminology choice, maybe I'd use "stud-pull adhesion strength" or "normal adhesion strength" for experiments that used stud pull method. Then reserve "peel-off strength" for experiments that use 90 degree peel.

3. Interesting about the pin width limitation of this conductor, making it incompatible with QFN. Definitely some small patterns can be created with direct writing and/or stencil-printing in the future. Many inks are solvent based. No change recommended here, just a comment.

4. Fig. R6 Chemical modification is interesting - seems it'll be worth trying other surface functionalizations in the future, such as silanes. I've never tried functionalizing a package, but presumably it could be done in a manufacturing setting if the cure time was low (couple minutes and compatible with a batch process, to avoid lowering total circuit throughput), and cure temperature was low (less than maybe 50C to avoid softening the reels used to dispense circuits). Why was this not included in the manuscript?

5. New Figure 1 is much clearer to this reviewer.

6. Appreciate the effort to use 4PP method instead of 2PP, and also trying laser cutting. While I do not think additional experiments are "necessary", my suggestion would be to try cutting with a high-resolution UV laser (~15 um spot size) at low power, over several iterations. It'll take longer than other methods, but doing multiple passes tends to reduce charring and other unwanted heat transfer outside of the cut region. Just want to see this work go far.

7. Re-iterating comment that most of the SI could be better explained. For example, S10 caption doesn't seem to have a conclusion, and it's barely mentioned in the main text as far as this reviewer can tell. Importantly, no information is given on how to interpret the Raman peaks or why 2900 cm⁻¹ was chosen. Related - this reviewer thinks that, at a minimum, all supplementary figures should be

mentioned in the text, and it appears S5, S11, S13 aren't.

8. Re-iterating comment that the robustness to shear should be explicitly tested. Not a deal-breaker, but this seems like an easy experiment, where you'd do the stud pull test but with a right-angle attachment instead of straight-line. Again, I can't recall other stretchable electronics papers studying this failure mode, but in my opinion, that's the whole point of this paper. The proposed techniques should advance the field to be able to actually do shear testing on stretchable electronics, instead of uniaxial tensile testing like the current state of the art.

9. While the article is quite readable, the grammar and word choice can be improved. Examples:

- Fig. 2a "Ultrasonication" is the process shown, while "Ultrasonic" is an adjective.

- Fig. 2 caption "Except where stated otherwise, polymers were prepared..." "Comparison of the tensile stress-strain curves..."

- Fig. 4 caption is still a little confusing. "Constituent". "The measured conductivity and yield of the ULPC over repeated recycling operations. Here, yield is defined as mass after recycling divided by initial mass."

- Methods "heat resistance resin on a stainless-steel half sphere" -- "heat resistant resin on a stainless-steel half sphere". Thermoforming process is in present tense, should be past.

10. Fig. 4 images are quite small. Hopefully Nat. Comm. editors can work with the authors to make this larger somehow.

Reviewer #3 (Remarks to the Author):

Compared to previous research on liquid metals, the primary innovation in this study lies in its weldability. However, the suggested title change to "Tough electronics enabled by small-molecule modulated interfacial assemblies" does not accurately reflect the main novelty of this article. This conductor is not inherently tough on its own, as it relies on a TPU substrate. Additionally, it does not constitute "electronics" in the traditional sense, as they did not make tough electronics and active components like transistors. To better align with the focus on soldering, a more appropriate title could be "Tough soldering for stretchable electronics by small-molecule modulated interfacial assemblies." The current title leads to confusion among readers and is not acceptable.

Regarding the suitability of this work for publication in Nature Communications, it should be focused on a stretchable solder, not the conductor itself. This material has some concerns, particularly in terms of its functionality as a conductor. This conductor cannot be patterned with high resolution (~50 um), which is essential for commercialized circuit board construction. Furthermore, it requires additional activation steps with peeling, a process that has been reported several times before. Among these, as a stretchable conductor, it can not be published in this journal.

1. Please focus on the soldering and avoid mentioning it as a conductor: mentioning it as a solderable electrode will be better. Please check this paper: <https://www.nature.com/articles/s41586-022-05579-z>

Other concerns include:

2. One more repeat: Please focus your materials as a 'self-solder'

3. Avoid indicating that the material is initially conductive through solvent evaporation, as it necessitates additional peeling steps.

4. The sedimentation test in the supplementary information requires a more detailed explanation and clarification.

I will re-review your work after proper revision.

Response to Reviewers' Comments

The Reviewer's comments are in black and revised texts are **highlighted**.

Reviewer #1

The authors have made detailed response to the reviewers' comments. And the manuscript has been improved. I have no further questions.

A: We appreciate the Reviewer for the careful review of our work.

Reviewer #2

Overall, the reviewers have greatly improved the manuscript, and addressed the vast majority of my comments. I do not have major concerns with the manuscript, but will still provide some comments below in the spirit of constructive criticism. This reviewer will look forward to reading this manuscript whenever/wherever it ends up being published, and replicating the results.

A: We are grateful to the Reviewer for the positive feedback and helpful comment. We have revised our work according to the Reviewer's comments. Point-by-point responses are attached below:

1. The "Our work" row in Table R1 - is the adhesion column supposed to have two values? It wasn't clear why two were listed for "Our work" but only one listed for the others.

A: We appreciate the Reviewer for bringing up this question. The presence of two values in the "Adhesion" column for the "Our work" row in Table R1 is due to the heterogeneous structure of our material. Specifically, there are different adhesion properties between the LM-rich side and the polymer-rich side (Supplementary Fig. 14). By including two values, we aim to illustrate the contrasting adhesion strengths between these two surfaces more effectively. Upon further consideration and to avoid confusion for readers, we have decided to only list the adhesion value of the LM-rich layer, which is the surface in contact with electronic components. This will provide a clearer representation of the relevant adhesion strength for practical applications. Related revisions have been made in Supplementary Table 4.

2. The adhesion experiments are still unclear to me. For example, the stud pull method

was mentioned in the SI but I can't figure out which figures used a 90-degree pull test and which used the stud pull test. In particular, Fig. 3 C is driving me crazy since I can't figure out which of the presented techniques was used. I would encourage the authors to double-check the revised manuscript for similar confusions - make sure each figure's technique is clearly mentioned and mention whether it's described in supplementary materials vs. main manuscript's "materials & methods" section. I would further encourage the authors to use clear terminology to reinforce the methods used. For instance, calling Fig. S23 "Peel off strength of zero-ohm resistors" seems like a mistake to this reviewer. I would assume Fig. S23 used stud pull method. For terminology choice, maybe I'd use "stud-pull adhesion strength" or "normal adhesion strength" for experiments that used stud pull method. Then reserve "peel-off strength" for experiments that use 90 degree peel.

A: We appreciate the Reviewer for the suggestion. We have made revisions to clarify the adhesion experiments in the manuscript. We now use "stud-pull strength" to refer to the adhesion strength measured using the stud-pull test, "peel-off strength" for the adhesion strength measured using the 90° peel-off test, and "shear strength" for the adhesion strength measured using the shear test. This will help readers better understand the specific techniques used in each experiment. We have also revised Supplementary Fig. 23, Fig. 24, and Fig. 25 and related descriptions in the Main Text. The details of 90° peel-off test and shear test have been added in the Supplementary Note.

3. Interesting about the pin width limitation of this conductor, making it incompatible with QFN. Definitely some small patterns can be created with direct writing and/or stencil-printing in the future. Many inks are solvent based. No change recommended here, just a comment.

A: We are grateful for the Reviewer's comment. We fully agree with you about the potential for direct writing and stencil-printing techniques to create smaller patterns in the future. We appreciate your input and will consider exploring these methods in our future research.

4. Fig. R6 Chemical modification is interesting - seems it'll be worth trying other surface functionalizations in the future, such as silanes. I've never tried functionalizing a package, but presumably it could be done in a manufacturing setting if the cure time was low (couple minutes and compatible with a batch process, to avoid lowering total circuit throughput), and cure temperature was low (less than maybe 50°C to avoid softening the reels used to dispense circuits). Why was this not included in the

manuscript?

A: We appreciate the Reviewer's comment on the potential of exploring other surface functionalization, such as silanes, and the considerations for implementing them in a manufacturing setting. While we agree that chemical modification holds promise for improving interface strength, we would like to clarify that our focus in this particular manuscript was on presenting the novel material design and its initial research findings. The inclusion of detailed discussions on the compatibility of thermal processes, cure time, and temperature would extend beyond the scope of this work and could dilute the impact of our primary research. Therefore, we did not include detailed discussions on the compatibility of thermal processes, cure time, and temperature in the manuscript. However, we have added these aspects in the supporting information as Supplementary Fig. 29 and mentioned them as part of our future targets in the conclusion section.

We value your understanding and recognition of the challenges involved in implementing such modifications in a manufacturing context. Your suggestion will certainly be taken into consideration for our future research endeavors.

5. New Figure 1 is much clearer to this reviewer.

A: We are grateful to the Reviewer for the positive feedback.

6. Appreciate the effort to use 4PP method instead of 2PP, and also trying laser cutting. While I do not think additional experiments are "necessary", my suggestion would be to try cutting with a high-resolution UV laser (~15 um spot size) at low power, over several iterations. It'll take longer than other methods, but doing multiple passes tends to reduce charring and other unwanted heat transfer outside of the cut region. Just want to see this work go far.

A: We are grateful to the Reviewer for the valuable suggestion. We truly appreciate your recognition of our efforts in utilizing the 4PP method instead of the 2PP method, as well as our attempt at laser cutting. We value your input and understand the potential advantages of using a high-resolution UV laser with a smaller spot size and performing multiple iterations at low power to reduce charring and unwanted heat transfer. Unfortunately, we currently do not have access to the specific equipment you mentioned within our constraints. However, we sincerely appreciate your recommendation, and we will keep it in mind for future research and experimentation.

7. Re-iterating comment that most of the SI could be better explained. For example, S10 caption doesn't seem to have a conclusion, and it's barely mentioned in the main

text as far as this reviewer can tell. Importantly, no information is given on how to interpret the Raman peaks or why 2900 cm⁻¹ was chosen. Related - this reviewer thinks that, at a minimum, all supplementary figures should be mentioned in the text, and it appears S5, S11, S13 aren't.

A: We are grateful to the Reviewer for the comment. In Supplementary Fig. 10, the distribution of the 2900 cm⁻¹ peaks, which are assigned to the asymmetric and symmetric stretches of the C-H in the PDMS backbone [1], indicates the distribution of polymer-rich domain. We have revised the caption of Supplementary Fig. 10 to provide a clearer conclusion. Additionally, we have checked all supplementary figures are appropriately mentioned in the text.

[1] Jayes, L.at, 2003 <https://pubs.acs.org/doi/10.1021/ac026012f>

8. Re-iterating comment that the robustness to shear should be explicitly tested. Not a deal-breaker, but this seems like an easy experiment, where you'd do the stud pull test but with a right-angle attachment instead of straight-line. Again, I can't recall other stretchable electronics papers studying this failure mode, but in my opinion, that's the whole point of this paper. The proposed techniques should advance the field to be able to actually do shear testing on stretchable electronics, instead of uniaxial tensile testing like the current state of the art.

A: We appreciate the Reviewer for the valuable feedback. We agree that it is important to conduct shear testing to further evaluate the robustness of our materials, as shear forces play a significant role in the performance of stretchable electronics.

To address this, we have conducted shear force testing using a standard method. During the test, samples consisting of ULPC bonded to Cu film substrates were securely clamped in a shear test fixture. Shear force was gradually applied until separation occurred. The results, including shear strength values, are included in Fig. R1. This test provides a quantitative measurement of the performance of ULPC under shear stress, which is relevant for the practical application of flexible circuits where shear forces are generated during bending or stretching.

Fig. R1 is added in Supplementary Information as Supplementary Fig. 15 to showcase the tensile curves for the shear strength test. The details of 90° peel-off test and shear test have been added in the Supplementary Note.

Fig. R1 The representative tensile curve for the shear strength test. The shear strength was measured according to the forces and the contact area of the Cu films and ULPC with 30 wt% LM content.

9. While the article is quite readable, the grammar and word choice can be improved. Examples:

- Fig. 2a "Ultrasonication" is the process shown, while "Ultrasonic" is an adjective.
- Fig. 2 caption "Except where stated otherwise, polymers were prepared..." "Comparison of the tensile stress-strain curves..."
- Fig. 4 caption is still a little confusing. "Constituent". "The measured conductivity and yield of the ULPC over repeated recycling operations. Here, yield is defined as mass after recycling divided by initial mass."
- Methods "heat resistance resin or a stainless-steel half sphere" -- "heat resistant resin on a stainless-steel half sphere". Thermoforming process is in present tense, should be past.

A: We appreciate the Reviewer for providing specific examples and suggestions for improvement. We have carefully reviewed the manuscript and made the necessary revisions to address any typos and errors in the manuscript.

10. Fig. 4 images are quite small. Hopefully Nat. Comm. editors can work with the authors to make this larger somehow.

A: We are thankful to the Reviewer for the helpful comment. We have increased the size of the images in Fig.4 to provide readers with a better view of the image.

Reviewer #3

Compared to previous research on liquid metals, the primary innovation in this study lies in its weldability. However, the suggested title change to "Tough electronics enabled by small-molecule modulated interfacial assemblies" does not accurately

reflect the main novelty of this article. This conductor is not inherently tough on its own, as it relies on a TPU substrate. Additionally, it does not constitute "electronics" in the traditional sense, as they did not make tough electronics and active components like transistors. To better align with the focus on soldering, a more appropriate title could be "Tough soldering for stretchable electronics by small-molecule modulated interfacial assemblies." The current title leads to confusion among readers and is not acceptable.

Regarding the suitability of this work for publication in Nature Communications, it should be focused on a stretchable solder, not the conductor itself. This material has some concerns, particularly in terms of its functionality as a conductor. This conductor cannot be patterned with high resolution (~50 μm), which is essential for commercialized circuit board construction. Furthermore, it requires additional activation steps with peeling, a process that has been reported several times before. Among these, as a stretchable conductor, it can not be published in this journal.

A: We appreciate the Reviewer for the valuable comments. We have taken your suggestions into consideration and have revised the title to "Tough soldering for stretchable electronics by small-molecule modulated interfacial assemblies" to better align with the focus on soldering. We have also made changes throughout the manuscript to highlight the soldering aspect and avoid emphasizing our material as conductors. Thank you for creatively pointing out the characteristics of self-soldering of our materials, which reflects the unique properties of our materials.

We would also like to emphasize the point that traditional printed circuit boards are not tailored or optimized for stretchable electronics applications. In recent years, there has been significant development in stretchable conductive materials based on LM. Our material not only demonstrates comparable stretchable conductive properties to other reported materials, but it also possesses the distinctive characteristic of self-soldering, which greatly facilitates seamless integration with electronic components. Additionally, we acknowledge the other limitations raised by the reviewer, and we are committed to addressing and improving upon them in our future work. Furthermore, we acknowledge that activation steps with peeling have been reported before, and we have clarified this in the manuscript.

We hope that these revisions address your concerns and demonstrate the suitability of our work for publication in Nature Communications. Point-by-point responses are attached below:

1. Please focus on the soldering and avoid mentioning it as a conductor: mentioning it as a solderable electrode will be better. Please check this paper: <https://www.nature.com/articles/s41586-022-05579-z>

A: We acknowledge the Reviewer for the suggestion. We have made the necessary changes throughout the manuscript to emphasize the soldering aspect and avoid highlighting our material as conductors. We have also cited the provided reference to provide additional context for the reader.

Other concerns include:

2. One more repeat: Please focus your materials as a ‘self-solder’

A: We appreciate the Reviewer for the comments. In the revised manuscript, we have placed greater emphasis on the self-soldering aspect and highlighted its significance in achieving stretchable electronics. We ensure that the materials are presented as a "self-solder" system throughout the manuscript.

3. Avoid indicating that the material is initially conductive through solvent evaporation, as it necessitates additional peeling steps.

A: We appreciate your comment regarding the initial conductivity through solvent evaporation and the subsequent peeling steps. In the revised manuscript, we have provided a clear description of the material's initial conductivity and its relationship to the peeling process - “With the compact assembly of LMP, peeling films of ULPC can generate sufficient stress to percolate particles for electrical activation (Supplementary Fig. 13)”. We ensure that readers understand the additional steps involved.

4. The sedimentation test in the supplementary information requires a more detailed explanation and clarification.

A: We acknowledge the Reviewer for bringing up concern regarding the sedimentation test in the supplementary information. The purpose of the sedimentation test was to demonstrate the role of UPy_{MC} modulators in promoting the aggregation and compact assembly of LMP, which is crucial for the peeling-induced activation process. In the absence of UPy_{MC} modulators, the LMP takes longer to settle to the bottom during the solvent evaporation process. This indicates that a smaller amount of LMP is settled at the bottom, resulting in a less compact assembly. As a result, after peeling off, the LMP particles cannot form an interconnected network, hindering the formation of a conductive path. We have provided a more detailed explanation and clarification of the

sedimentation test in the revised supplementary information. We hope that the additional information provided will enhance the understanding of the experiment.

I will re-review your work after proper revision.

A: We are grateful to the Reviewer and look forward to your re-review of the revised manuscript.

REVIEWER COMMENTS

Reviewer #2 (Remarks to the Author):

The reviewers have improved the manuscript again, and their responses were appropriate. Reviewer 3 has brought up some good points, and I can agree that "soldering" is an accurate term for one of this conductor's functions. The use of this soldering then enables tough electronics (in the whole-circuit sense of the term; not individual components as Reviewer 3 points out). I personally am okay if the authors want to call their material a solder or a conductor, as it serves both functions in different capacities... but do agree with Reviewer 3 that the primary contributions of this paper are in the soldering (adhesion, bonding...) function.

Couple odd "almost complete" aspects of the revisions...

- Supplementary Fig. 5 seems to still not be mentioned in the main text.
- Only one shear test was completed. It takes only 20% more effort to do three samples compared to 1! (Not to mention that the test would be more useful to others when presented next to a control. The presented test is still useful to get a ballpark range - 2 MPa is like a strong silicone adhesive, but less than many epoxies.)

Reviewer #3 (Remarks to the Author):

The manuscript has become more suitable for Nature Communications. However, there are still some inadequacies in the underlying mechanism concerning soldering:

1. Figure 2 or 3 should be modified to offer a more comprehensive illustration of the mechanisms related to self-soldering. Presently, there are no figures that depict the soldering mechanism, and I look forward to conducting a thorough review of the revised version once these enhancements are incorporated.
2. Figure 1f is not relevant to tough soldering. It should be replaced with an illustration of the soldering of electronic chips. The figure should contain multiple chips to demonstrate the ease of the soldering process. This manuscript may not be suitable for publication in Nature Communications if the author claims it represents a printable stretchable conductor.

Response to Reviewers' Comments

The Reviewer's comments are in black and revised texts are highlighted.

Reviewer #2

The reviewers have improved the manuscript again, and their responses were appropriate. Reviewer 3 has brought up some good points, and I can agree that "soldering" is an accurate term for one of this conductor's functions. The use of this soldering then enables tough electronics (in the whole-circuit sense of the term; not individual components as Reviewer 3 points out). I personally am okay if the authors want to call their material a solder or a conductor, as it serves both functions in different capacities... but do agree with Reviewer 3 that the primary contributions of this paper are in the soldering (adhesion, bonding...) function.

A: We are grateful to the Reviewer for the positive feedback and helpful comment. We have revised our work according to the Reviewer's comments. Point-by-point responses are attached below:

Couple odd "almost complete" aspects of the revisions...

- Supplementary Fig. 5 seems to still not be mentioned in the main text.

A: We appreciate the Reviewer for the comment. We have added descriptions about Supplementary Fig. 5 in the Main Text as follows: EDS analysis revealed a homogeneous distribution of carbon elements surrounding the LMP surface, providing further evidence of the uniform wrapping of UPy_{MC} around the LMP surface (Supplementary Fig. 5).

- Only one shear test was completed. It takes only 20% more effort to do three samples compared to 1! (Not to mention that the test would be more useful to others when presented next to a control. The presented test is still useful to get a ballpark range - 2 MPa is like a strong silicone adhesive, but less than many epoxies.)

A: We appreciate the Reviewer for the suggestion to perform multiple tests and include a control group for comparison. We have added two duplicated samples and two control samples of commercial adhesive tape in the shear test (Fig. R1). We believe this would facilitate better comparisons with other adhesive materials.

Fig. R1 Representative tensile curves for shear strength test. The shear strength was measured according to the forces and the contact area of the Cu films and samples. Samples 1-3 are duplicated ULPC samples with 30 wt% LM content. 3M VHB tape 5952 and 3M double-sided tape 9448A are used as control. From the result, the maximum shear strength of the ULPC sample was 2.1 ± 0.08 MPa, which shows competitive adhesion performance with double-sided tape and was superior to VHB tape.

Reviewer #3

The manuscript has become more suitable for Nature Communications. However, there are still some inadequacies in the underlying mechanism concerning soldering:

A: We are thankful for the Reviewer's positive feedback on the revised manuscript and continued engagement with the underlying mechanism concerning soldering. We have provided additional explanations and schematics to enhance the understanding of the soldering process. Point-by-point responses are attached below:

1. Figure 2 or 3 should be modified to offer a more comprehensive illustration of the mechanisms related to self-soldering. Presently, there are no figures that depict the soldering mechanism, and I look forward to conducting a thorough review of the revised version once these enhancements are incorporated.

A: We are grateful to the Reviewer for the suggestion. In response, we have made modifications to enhance the visualization of the soldering mechanism. Specifically, we have revised Fig. 3d to better depict the self-soldering process, as shown in Fig. R2. During the soldering process, the ULPC deforms to better wrap the mounted resistor after thermal processing. Additionally, the heterogeneous structure of the ULPC enables the formation of a connection that possesses both stable conductivity and adhesion with electrical components. We believe these modifications will provide a more

comprehensive illustration of the mechanisms related to self-soldering.

Fig. R2 Schematic illustration and optical images of the side view of the soldering process of ULPC-TPU with a zero-ohm resistor, scale bar, 1 mm.

2. Figure 1f is not relevant to tough soldering. It should be replaced with an illustration of the soldering of electronic chips. The figure should contain multiple chips to demonstrate the ease of the soldering process. This manuscript may not be suitable for publication in Nature Communications if the author claims it represents a printable stretchable conductor.

A: We appreciate the Reviewer for the suggestion. We have revised Fig. 1f to highlight the ease of the soldering process as now shown in Fig. R3. We have revised the relevant description in the manuscript to better align with this new illustration. We also want to highlight the solder's strong adhesion during the thermoforming process. This adhesion is crucial in resisting the significant shear force between flexible conductors and rigid electronic devices while maintaining conductivity under high temperatures and stretching.

Fig. R3 Schematic illustration of the soldering and thermoformed processes.

REVIEWER COMMENTS

Reviewer #2 (Remarks to the Author):

Another great round of improvements. The edited figures, especially shear force R1 (Fig. S15) and soldering R2 (Fig. 3) are much appreciated. I have no further substantial comments, but provide a couple quick suggestions that are really just typesetting/copyedits on the current revision material:

- Fig. R3 (Fig. 1) maybe "2D planar assembly" or "2D component placement" or "2D assembly". I just think there's a missing action verb

- Fig. 1 caption: "Schematic of the soldering and thermoforming process."

- P.4 Couple comma/verb tense/style choices I think will improve readability:

"The dynamic interactions within the ULPC enable a stable interconnect between both electronic components and TPU substrate to withstand the significant shear stress generated during the thermoformed process. Notably, the circuit depicted in Fig. 1f exhibits exceptional stability and robustness without requiring encapsulation, which simplifies component substitution and circuit recyclability."

"The dynamic interactions within the ULPC enable a stable interconnect between both electronic components and TPU substrate, to withstand the significant shear stress generated during the thermoforming process. Notably, the circuit exhibits exceptional stability and robustness without requiring encapsulation, which simplifies component substitution and circuit recyclability (Fig. 1F)." or, changing the location of the figure reference

"The dynamic interactions within the ULPC enable a stable interconnect between both electronic components and TPU substrate, to withstand the significant shear stress generated during the thermoforming process. Notably, the circuit (Fig. 1F) exhibits exceptional stability and robustness without requiring encapsulation, which simplifies component substitution and circuit recyclability."

Reviewer #3 (Remarks to the Author):

In general, the changed topic of revised manuscript seems suitable for Nature Communications. However, I believe that some further revisions are necessary to enhance the quality and relevance of the paper. Specifically, I recommend the following:

1. Please provide more detailed explanation of the soldering mechanism involving your polymer. Is dynamic bonding the only crucial factor? If so, it would be important to add references to previous works that discuss this mechanism.

2. I suggest adding one more demonstration involving two different conductors (e.g., conventional metals or conducting materials like CNT) with the introduced materials to showcase the validity of using them as solder. You can refer to this paper for examples:

<https://www.nature.com/articles/s41586-022-05579-z>.

3. I recommend enriching the Introduction section with additional references to cover the latest developments in the research area. For instance,

- robust soldering techniques: <https://www.nature.com/articles/s41586-022-05579-z>.

- liquid metal and polymer composites: <https://www.nature.com/articles/s41563-020-00863-7> and <https://www.nature.com/articles/s41467-022-30427-z>.

Response to Reviewers' Comments

The Reviewer's comments are in black and revised texts are **highlighted**.

Reviewer #2

Another great round of improvements. The edited figures, especially shear force R1 (Fig. S15) and soldering R2 (Fig. 3) are much appreciated. I have no further substantial comments, but provide a couple quick suggestions that are really just typesetting/copyedits on the current revision material:

A: We appreciate the Reviewers for the positive feedback and careful reviews of our work. Point-by-point responses are attached below:

- Fig. R3 (Fig. 1) maybe "2D planar assembly" or "2D component placement" or "2D assembly". I just think there's a missing action verb

A: We are grateful to the Reviewer for the valuable suggestion. We have changed "2D planar" to "2D planar assembly" to provide a clearer and more accurate description of Fig. 1.

- Fig. 1 caption: "Schematic of the soldering and thermoforming process."

A: We thank the Reviewer for the valuable suggestion. We have revised the Fig. 1 caption as you recommended.

- P.4 Couple comma/verb tense/style choices I think will improve readability:

"The dynamic interactions within the ULPC enable a stable interconnect between both electronic components and TPU substrate to withstand the significant shear stress generated during the thermoformed process. Notably, the circuit depicted in Fig. 1f exhibits exceptional stability and robustness without requiring encapsulation, which simplifies component substitution and circuit recyclability."

 "The dynamic interactions within the ULPC enable a stable interconnect between both electronic components and TPU substrate, to withstand the significant shear stress generated during the thermoforming process. Notably, the circuit exhibits exceptional stability and robustness without requiring encapsulation, which simplifies component substitution and circuit recyclability (Fig. 1F)."

or, changing the location of the figure reference

"The dynamic interactions within the ULPC enable a stable interconnect between both electronic components and TPU substrate, to withstand the significant shear stress generated during the thermoforming process. Notably, the circuit (Fig. 1F) exhibits exceptional stability and robustness without requiring encapsulation, which simplifies component substitution and circuit recyclability."

A: We are grateful to the Reviewer for the valuable suggestion. We have revised the description as "The dynamic interactions within the ULPC enable a stable interconnect between both electronic components and TPU substrate, to withstand the significant shear stress generated during the thermoforming process. Notably, the circuit (Fig. 1f) exhibits exceptional stability and robustness without requiring encapsulation, which simplifies component substitution and circuit recyclability."

Reviewer #3

In general, the changed topic of revised manuscript seems suitable for Nature Communications. However, I believe that some further revisions are necessary to enhance the quality and relevance of the paper. Specifically, I recommend the following:

A: We appreciate the Reviewers for the feedback on our work. Point-by-point responses are attached below:

1. Please provide more detailed explanation of the soldering mechanism involving your polymer. Is dynamic bonding the only crucial factor? If so, it would be important to add references to previous works that discuss this mechanism.

A: We thank the Reviewer for the valuable feedback on our revised manuscript. For an explanation of the soldering mechanism of our materials, we have included Fig. 1 and Fig. 3 to highlight their characteristics. In our manuscript, we emphasize the unique heterogeneously structured ULPC consisting of liquid metal-rich domains and polymer-rich domains, which is beneficial for interfacial contact with high conductivity and high adhesion respectively. During the soldering process, the polymer composite deforms to better wrap the mounted components. The ULPC leverages dynamic interactions to establish a stable interconnect between both electronic components and the TPU substrate. Specifically, quadruple hydrogen bonds are formed at the interface, contributing to high adhesion. These interactions enable a reliable connection, ensuring the seamless integration of electronic elements within stretchable devices. To support

this explanation, we added some references in the "Interface-Dependent Electromechanical Properties of ULPC" section of the manuscript as follows:

-- "Since the LM content in the hybrid interface contributes to the conductivity and the polymer content contributes to the interfacial adhesion, such unique hybrid interface is beneficial for interfacial contact with high conductivity and high adhesion [1]".

-- "The stable adhesion relies on molecular bond exchange and recombination when the ULPC is brought into contact with electrical components [2]".

-- "Fig. 3d shows schematics and optical images of a resistor-mounted ULPC-TPU, where the polymer composite deforms to better wrap the mounted resistor after thermal processing [3]".

[1] <https://www.nature.com/articles/s41586-022-05579-z>

[2] <https://pubs.acs.org/doi/epdf/10.1021/jacs.8b01682>

[3] <https://www.nature.com/articles/ncomms1980>

We believe that these revisions address your concerns and provide the requested clarification on the soldering mechanism involving our composites. Thank you again for your valuable input, which has undoubtedly improved the quality of our manuscript.

2. I suggest adding one more demonstration involving two different conductors (e.g., conventional metals or conducting materials like CNT) with the introduced materials to showcase the validity of using them as solder. You can refer to this paper for examples: "<https://www.nature.com/articles/s41586-022-05579-z>"

A: We are grateful to the Reviewer for the advice to add additional demonstrations showcasing the validity of using our materials as solder. Upon further consideration, we believe that including our materials with conventional metals or conducting materials like CNT would divert the focus of our article and dilute the core message we aimed to convey. However, we acknowledge the importance of demonstrating our material's compatibility with other conductive materials.

In response to this concern, we have conducted further experiments and included them in our revised manuscript. Specifically, we have validated the feasibility of our materials with bGaIn [4] (biphasic gallium-indium) and O-GaIn [5] (oxidized Gallium-Indium), which are representative liquid metals. The results of these experiments show that there is no significant difference between using our materials with EGaIn and these different liquid metals (Fig. R1). Furthermore, the resulting composites exhibit high conductivity after the soldering process, as evidenced by the well-lit LED. These

findings highlight the versatility and potential of our approach in utilizing different liquid metals to achieve the desired properties as solder. Fig. R1 is added in supplementary information as Supplementary Fig. 30.

[4] <https://www.nature.com/articles/s41563-021-00921-8>

[5] <https://onlinelibrary.wiley.com/doi/abs/10.1002/adfm.201907063>

Fig. R1 Optical images of prepared **a**, O-GaIn and **b**, bGaIn polymer composites and they integrated with LED chip by the soldering process.

3. I recommend enriching the Introduction section with additional references to cover the latest developments in the research area. For instance,

- robust soldering techniques: <https://www.nature.com/articles/s41586-022-05579-z>.

- liquid metal and polymer composites: <https://www.nature.com/articles/s41563-020-00863-7> and <https://www.nature.com/articles/s41467-022-30427-z>

A: We appreciate the Reviewer's suggestion to enrich the Introduction section with additional references. We have carefully considered the recommendations and made the following changes:

We have included the Nature paper on robust soldering techniques (ref. 9) in the Introduction section of the manuscript. Additionally, we have cited the Nature Communications paper (<https://www.nature.com/articles/s41467-022-30427-z>) for the chemical modification of liquid metal particles, which aligns better with the focus of our study. However, after careful consideration, we have decided not to include the Nature Materials paper on hydrogen-doped conductive LM (<https://www.nature.com/articles/s41563-020-00863-7>) in the Introduction section as it is more relevant to conductive ink rather than solder, and we believe it may not provide significant

additional value to the context of our study.

We believe that these modifications effectively address the Reviewer's suggestion to cover the latest developments in the research area while maintaining the focus of our manuscript.

REVIEWER COMMENTS

Reviewer #3 (Remarks to the Author):

Regarding Q2, I believe there might be a misunderstanding regarding my point. I still believe that a demonstration showcasing the stretchable solder properties of your materials is necessary. This demonstration could involve metal-solder-metal or conductor-solder-conductor soldering. Please refer to the provided reference for further clarification.

<https://www.science.org/doi/10.1126/sciadv.abh0171>

Once a suitable demonstration is provided, I will agree to accept the paper.

Response to Reviewers' Comments

The Reviewer's comments are in black and revised texts are **highlighted**.

Reviewer #3

Regarding Q2, I believe there might be a misunderstanding regarding my point. I still believe that a demonstration showcasing the stretchable solder properties of your materials is necessary. This demonstration could involve metal-solder-metal or conductor-solder-conductor soldering. Please refer to the provided reference for further clarification. <https://www.science.org/doi/10.1126/sciadv.abh0171>

Once a suitable demonstration is provided, I will agree to accept the paper.

A: We appreciate the Reviewer for the feedback and clarifying the point in Q2. To address this concern, we have carefully reviewed the provided reference (<https://www.science.org/doi/10.1126/sciadv.abh0171>) for further clarification.

To address this concern, we have incorporated a suitable demonstration highlighting the stretchable solder properties of our materials in the manuscript (Fig. R1). We have added a paragraph to discuss this demonstration:

"In addition, we have expanded the application of our solder with other circuits (rigid, flexible, or stretchable) as a stretchable connector. By soldering ULPC to flat flexible cables and flexible silver conductors, we have successfully formed soft-rigid and soft-soft stretchable connections (See Supplementary Fig. 30 for details). These substrates are compatible with printed circuit board manufacturing, allowing the full utilization of the performance of chips and electronic components. This demonstration showcases the potential of our solder in building stretchable hybrid devices with different functionality and complexity."

Fig. R1 is added in the Supplementary Information as Supplementary Fig. 30. We have also cited the provided reference as ref.10 to provide additional context for the reader.

Thank you for your valuable feedback, and we believe that the revised manuscript now adequately addresses your concerns.

Fig. R1 An 8-channel capacitive sensor array assembled by ULPC solder connection. **a**, Schematic of the application of our solder with other circuits (rigid, flexible, or stretchable) as a stretchable connector. **b**, Photograph of a ULPC-TPU 8-channel capacitive sensor array, consisting of three parts: stretchable electrodes, flat flexible cables/ flexible silver conductors (700 μm linewidth and 1 mm pitch), and the printed circuit board. **c**, Capacitive signal from 8 channels obtained from ULPC electrodes when we touch the capacitive touch pad of each channel in turn. The stable and low-impedance connection interface ensures accurate output signals reflecting the capacitive change of each channel with low crosstalk and noise.

REVIEWERS' COMMENTS

Reviewer #3 (Remarks to the Author):

Now, I agree to accept the publication of this article. I hope the authors also agree that changing the topic and making further revisions has improved the overall quality and scientific importance of the article. Congratulations!

Response to Reviewers' Comments

Reviewer #3

Now, I agree to accept the publication of this article. I hope the authors also agree that changing the topic and making further revisions has improved the overall quality and scientific importance of the article. Congratulations!

A: We are pleased that our revisions have satisfactorily addressed the concerns raised by the Reviewer. We greatly appreciate the valuable comments provided by the Reviewer, which helped us to refine our work and enhance the quality of the manuscript.